# Non-Invertible T-duality at Any Radius via Non-Compact SymTFT

Riccardo Argurio,[a] Andrés Collinucci,[a] Giovanni Galati,[a]
Ondrej Hulik,[b] and Elise Paznokas[a]

[a]*Physique Théorique et Mathématique and International Solvay Institutes*

*Université Libre de Bruxelles, C.P. 231, 1050 Brussels, Belgium*

[b]*Institute for Mathematics Ruprecht-Karls-Universitat Heidelberg,*

*69120 Heidelberg, Germany*

**Abstract**

We extend the construction of the T-duality symmetry for the 2d compact boson to arbitrary values of the radius by including topological manipulations such as gauging continuous symmetries with flat connections. We show that the entire circle branch of the $c = 1$ conformal manifold can be generated using these manipulations, resulting in a non-invertible T-duality symmetry when the gauging sends the radius to its inverse value. Using the recently proposed symmetry TFT describing continuous global symmetries of the boundary theory, we identify the topological operator corresponding to these new T-duality symmetries as an open condensation defect of the bulk theory, constructed by (higher) gauging an $\mathbb{R}$ subgroup of the bulk global symmetries. Notably, when the boundary theory is the compact boson with a rational square radius, this operator reduces to the familiar T-duality defect described by a Tambara-Yamagami fusion category. This construction thus naturally includes all possible discrete T-duality symmetries of the theory in a unified way.

# 1 Introduction

The work of [1] has put forward that symmetries are implemented in a relativistic quantum field theory (QFT) by topological defects. In a way, this tells us that symmetries of a QFT are related to its topological sector. In turn, the latter is naturally encoded in a topological QFT (TQFT), actually in one dimension higher as in the symmetry TQFT (SymTFT) paradigm [2–7]. This raises the question of what type of TQFTs one should be dealing with. A basic remark is that many QFTs have continuous symmetries, and hence a continuous infinity of topological defects. While most examples of SymTFTs are based on TQFTs with a finite number of defects,[1] it is clear that a generalization to TQFTs with an infinite number

---

[1]In this context, the SymTFT is limited to describing only the finite symmetries of the boundary QFT. However, it has proven to be highly effective in formalizing a wide range of phenomena, including the study of non-invertible symmetries (see e.g. [8–12]) together with the characterization of their anomalies (e.g. [13–18]), the study of gapped (e.g. [19, 20]) and gapless (e.g. [21, 22]) phases with categorical symmetries, and the

of defects is both needed and natural.

In the recent works [25–27] it was realized that a set of such TQFTs consists in BF theories with gauge connections taking values in non-compact gauge groups.[2] Since we will stick to abelian symmetries, the gauge group of interest will be $\mathbb{R}$. We will usually refer to these theories as non-compact TQFTs. In this context, it is natural to ask whether, within this framework, one can identify and describe more exotic types of generalized symmetries, whose elements are labeled by continuous rather than discrete parameters. Such symmetries would extend beyond the conventional framework of fusion categories, which typically assumes a finite number of simple objects.[3]

As a paradigmatic case of a QFT with continuous symmetries, in this paper we consider the theory of a compact boson in 2d. This is actually a CFT, indeed probably the most studied one (see e.g. [37]), while also exhibiting a range of subtle features. This theory has two $U(1)$ (0-form) symmetries, usually labeled as 'momentum' and 'winding'. Its SymTFT was described in [25] (see also [38, 39]), and is simply a 3d BF theory with two gauge fields in $\mathbb{R}$. An interesting observation is that the radius of the compact scalar can naturally be encoded in the topological boundary conditions of the symTFT [25, 38]. Since the TQFT is non-compact, there is indeed a continuous choice of boundary conditions. As a consequence, a rescaling of the radius can be considered as a change in the topological boundary conditions of the SymTFT. In the SymTFT dictionary, a change in the topological boundary conditions is usually interpreted as a topological manipulation in the physical theory. In the following, we show that this is indeed the case.

A key concept that goes hand in hand with non-compact TQFTs is the one of gauging a continuous group with flat connections. In this approach we effectively gauge the symmetry, ensuring that the field strength of the gauge field remains identically zero, and we perform the path integral only over flat connections. For instance, given the compact scalar, we can gauge its winding symmetry. This operation removes all the winding states, and renders the vertex operators with any non-quantized momentum genuine. Namely, the scalar is decompactified. Such a theory has now a non-compact $\mathbb{R}$ momentum 0-form symmetry. In a second step, we can further gauge a $\mathbb{Z}$ subgroup of $\mathbb{R}$. This new operation restores a quantization of the momentum of the vertex operators, with a new radius determined by the periodicity in choosing the emedding $\mathbb{Z} \subset \mathbb{R}$. Notice that we are generating the entire circle branch of the $c = 1$ conformal manifold with topological manipulations. If we also include

---

study of solitonic particles and their scattering properties (e.g. [23, 24]).

[2]See [28] for an alternative proposal to describe continuous symmetry from the perspective of a bulk non-topological theory.

[3]For some examples of constructions involving continuous non-invertible symmetries, see [29–36].

the gauging of charge conjugation, we actually generate the entire connected component of the conformal manifold.

Seeing an arbitrary rescaling of the radius as a topological manipulation, we are now in a position to define symmetry defects that combine T-duality of the compact boson with a rescaling from $R$ to $1/R$. The general procedure to construct topological duality lines for this type of theories is to perform the topological manipulation on half-space [40, 41]. Since the two sides are quantum mechanically the same theory, the interface separating the gauged and ungauged theory is actually a topological operator of the theory, thus generating a global symmetry. The existence of such duality defects, which are non-invertible outside the self-dual radius, has been recognized for a long time in theories with a rational squared radius [42]. In these cases, the conformal field theory (CFT) is rational.[4] For these specific radii, the definition of the duality line involves gauging only a finite $\mathbb{Z}_N$ subgroup of $U(1) \times U(1)$ forming the so-called Tambara-Yamagami fusion category [45]. We argue that using non-compact TQFTs, we can extend this understanding to any value of the radius. Following the same strategy as in the rational case, we show that the non-invertible symmetry defects of the compact scalar theory arise in the SymTFT as condensation defects [46, 47] of a specific $\mathbb{R}$ subgroup of the $\mathbb{R} \times \mathbb{R}$ global symmetry.[5] In other words, we need to higher gauge on a codimension one surface a non-compact, continuous symmetry, with flat connections. We show that, similarly to their discrete counterpart [8, 9], these defects are invertible as long as they are closed, whereas when open, their boundary gives rise to a non-invertible defect, the sought-for T-duality symmetry defect of the boundary theory.[6] We show that, by varying the topological boundary condition, the latter condensate of the non-compact TQFT nicely reduces to all possible duality defects of the boundary CFT.

The paper is structured as follows. In Section 2 we discuss the SymTFT for the 2d compact boson, emphasizing the boundary conditions at the topological and at the physical boundaries. In Section 3 we show how changing the radius is obtained by performing two successive gaugings, from a 2d perspective, and discuss how the non-invertible duality symmetry acts on vertex operators for generic radius. In Section 4 we build the condensation defects in the SymTFT, by explicitly taking continuous sums (integrals, that is) of

---

[4]However, categorical symmetries are not exclusive to rational CFTs. For examples of non-rational CFTs with categorical symmetries, see e.g. [42–44].

[5]For a previous discussion on the interpretation of T-duality (and its non-Abelian generalization) from a SymTFT perspective, see [48].

[6]To make those twist defects genuine, one should gauge the invertible symmetry generated by the closed condensation defects in the bulk. The gauged SymTFT now enjoys genuine topological operators describing the bulk dual of the T-duality defect [8, 9]. However the information we will need in the present work is nicely encoded in the ungauged SymTFT, by looking at the condensation defects and their boundaries.

line defects. We then specialize to the T-duality symmetry defect and check that it acts as expected when open. In Section 5 we give an analogous treatment for the known case of a SymTFT based on the usual compact TQFT, namely for the scalar at rational radius. The purpose is twofold: to demonstrate our approach in a case where the TQFT is finite, and also to discuss how the general case in the continuous case reduces to the known case when the radius takes rational values. In the Appendices, we present both some background material, and further details of our construction.

## 2 The Symmetry TFT

In this section we review basics of three dimensional BF theory as a SymTFT for 2-dimensional theories with $U(1) \times U(1)$ global symmetries. In particular, as proposed in [25,26], these symmetries are captured by a 3d TFT constructed out of two $\mathbb{R}$-valued 1-form gauge connections $b^{\pm}$ (see Appendix A for some remarks on the distinctions between $\mathbb{R}$- and $U(1)$-valued gauge fields) with action

$$S = \frac{iN}{2\pi} \int_{X_3} b^+ db^- \,, \tag{2.1}$$

and gauge transformations

$$b^{\pm} \to b^{\pm} + d\lambda^{\pm} \,. \tag{2.2}$$

Since $b^{\pm}$ are $\mathbb{R}$-valued gauge fields, we can perform a field redefinition $b^{\pm} \to \alpha b^{\pm}$ for any $\alpha \in \mathbb{R}$, without spoiling any quantization conditions. Therefore, we can set $N = 1$ without any loss of generality. This is a crucial difference with respect to the case in which $B^{\pm}$ are $U(1)$-valued gauge fields, where $N$ is quantized and the theory is quantum mechanically equivalent to a $\mathbb{Z}_N$ gauge theory [49]. In the following, we will highlight some of the differences between these two setups. Throughout this work, we will conventionally denote with capital letters $U(1)$-valued gauge fields while with lowercase letters we denote $\mathbb{R}$-valued gauge fields.

The equations of motion imply that $db^{\pm} = 0$, thus the only non-trivial gauge invariant observables are line operators constructed out of the holonomies of the gauge fields

$$U_x(\gamma) = \exp\left(ix \int_{\gamma} b^+\right) \quad, \quad V_y(\gamma) = \exp\left(iy \int_{\gamma} b^-\right), \tag{2.3}$$

where the dependence on $\gamma$ is purely topological and $x, y$ are real parameters. Notice that this TQFT enjoys an infinite set of line operators continuously parametrized by $x$ and $y$. We refer to these as non-compact TQFTs, owing to the non-compact spectrum of operators.

From the equations of motion we can derive braiding relations between line operators:

$$\langle U_x(\gamma) V_y(\gamma') \rangle = \exp\left(2\pi i \, xy \, \text{Link}(\gamma, \gamma')\right). \tag{2.4}$$

Thus the $\mathbb{R}$-valued BF theory has an $\mathbb{R} \times \mathbb{R}$ 1-form symmetry generated by $U_x, V_y$ with a non-trivial 't Hooft anomaly, captured by the braiding (2.4).[7]

When placed in a 3d manifold with boundaries, we need to specify boundary conditions for $b^\pm$. The manifold we will consider is a slab $X_3 = \Sigma_2 \times I$, where $\Sigma_2$ is a 2d manifold and $I$ is an interval, so that $\partial X_3 = \Sigma_2 \cup \overline{\Sigma}_2$.[8] On one of the two boundaries, $\Sigma_2$, we impose dynamical boundary conditions while on the other one, $\overline{\Sigma}_2$, we impose topological boundary conditions that specify the symmetry content of the theory.

## 2.1   Topological Boundary Conditions

Let us start by analyzing the full set of topological boundary conditions. We take the approach of coupling the theory to topological edge modes living on the boundary, which cancel the gauge variation coming from the bulk theory. Equivalently, topological boundary conditions of 3d Abelian TQFTs are in one-to-one correspondence with Lagrangian subgroups (see e.g. [51]), i.e. maximal sets of bulk line operators that have trivial braiding among themselves.

Performing a gauge transformation of the action (2.1), and focusing on the boundary at $\overline{\Sigma}_2$, we get

$$\delta S = \frac{\mathrm{i}}{2\pi} \int_{\overline{\Sigma}_2} \lambda^+ db^- \,. \tag{2.5}$$

To ensure gauge invariance, we can add topological boundary edge modes. The bulk/boundary system is now described by the action

$$S_{3d/2d} = S + \frac{\mathrm{i}R}{2\pi} \int_{\overline{\Sigma}_2} \Phi \, db^- \,, \tag{2.6}$$

where $\Phi$ is a *compact* scalar (i.e. $U(1)$-valued such that $\Phi \sim \Phi + 2\pi$) subject to the linear gauge transformation

$$\Phi \to \Phi - R^{-1}\lambda^+ \,, \tag{2.7}$$

and $R$ is an arbitrary real constant parametrizing a continuous family of boundary conditions.

The equations of motion on $\overline{\Sigma}_2$ and the sum over fluxes $\int_{\gamma \subset \overline{\Sigma}_2} d\Phi \in 2\pi\mathbb{Z}$ impose

$$b^+|_{\overline{\Sigma}_2} = -Rd\Phi \qquad , \qquad \int_{\gamma \subset \overline{\Sigma}_2} b^- \in 2\pi R^{-1}\mathbb{Z} \,. \tag{2.8}$$

---

[7]This anomaly should not be confused with a 't Hooft anomaly of the boundary QFT. The latter is naturally embedded in the SymTFT framework as the absence of a topological boundary condition corresponding to the global variant with the gauged symmetry [13, 16, 17, 50].

[8]We denote $\overline{\Sigma}_2$ the surface with opposite orientation with respect to $\Sigma_2$. We can also write $\overline{\Sigma}_2 = -\Sigma_2$.

Thus, they are boundary conditions trivializing the lines $U_{\frac{1}{R}n}, V_{Rw}$ with $n, w \in \mathbb{Z}$. This set of lines is a Lagrangian algebra of the bulk TQFT [25, 38]. By varying $R$, one generates all such Lagrangian algebras. The nontrivial boundary lines are now

$$U_x(\gamma), V_y(\gamma) \quad , \quad x \sim x + \frac{1}{R} \, , \, y \sim y + R \, . \tag{2.9}$$

Therefore, they generate a $U(1) \times U(1)$ 0-form global symmetry of the boundary theory. Notice that the limits $R \to 0, \infty$ correspond to Dirichlet boundary conditions for $b^{\pm}$ respectively, such that the boundary symmetry is $\mathbb{R}$.

## 2.2 Physical Boundary and Slab Compactification

We now want to discuss the non-topological boundary conditions imposed at the other boundary $\Sigma_2$ of the 3d slab manifold $X_3$. Generically, this boundary can host any 2d QFT with a $U(1) \times U(1)$ 0-form global symmetry. One of the simplest 2d theories of this kind is the $c = 1$ compact boson, which is known to arise by imposing conformal boundary conditions [52][9]

$$b^-|_{\Sigma_2} = \mathrm{i} \star_2 b^+|_{\Sigma_2} \, . \tag{2.10}$$

These boundary conditions[10] can be implemented by the following bulk/boundary action

$$S_{2d/3d} = S + \frac{1}{4\pi} \int_{\Sigma_2} b^- \wedge \star_2 b^- \, . \tag{2.11}$$

Note that we have used the leftover rescaling $b^{\pm} \to \beta^{\mp 1} b^{\pm}$ (with $\beta \in \mathbb{R}$) to fix the coefficient of the boundary action to its most convenient value, hence eliminating an otherwise arbitrary constant.

After setting the topological and physical boundary conditions, the symmetry TFT dictionary should produce a well-defined theory with a given global symmetry. This is done by slab compactification, i.e. by collapsing together the two boundaries, achieved by evaluating the full action on-shell:

$$S_{2d/3d/2d} = \frac{\mathrm{i}R}{2\pi} \int_{\Sigma_2} d\Phi \, \wedge b^- + \frac{1}{4\pi} \int_{\Sigma_2} b^- \wedge \star_2 b^- \, , \tag{2.12}$$

---

[9]Notice that the boundary conditions (2.10) do not depend on extra boundary edge modes. Thus the boundary theory is completely determined by the bulk TQFT and its topological boundary condition. This idea was recently studied in [38, 39] and interpreted as a toy model of the holographic principle.

[10]The gauge transformations of $b^{\pm}$ have to be restricted at $\Sigma_2$ in order to be compatible with the boundary conditions.

where we have used the fact that $\overline{\Sigma}_2 = -\Sigma_2$. Since the two boundaries have been super-imposed, we can enforce the topological and conformal boundary conditions (2.8), (2.10) at the same time, obtaining $b^- = -iR \star_2 d\Phi$ on $\Sigma_2$. We finally get

$$S_{2d} = \frac{R^2}{4\pi} \int_{\Sigma_2} d\Phi \wedge \star_2 d\Phi \ . \tag{2.13}$$

This is exactly the action describing a compact boson with radius $R$.

We can interpret a shift in boundary conditions specified by $R$ to ones specified by $R'$ as a topological manipulation sending $R$ to $R'$. When $R'/R = q/p \in \mathbb{Q}$ (with $\gcd(p,q) = 1$), this is the gauging of the subgroup $\mathbb{Z}_p \times \mathbb{Z}_q$ of the momentum and winding symmetry $U(1)_m \times U(1)_w$. For generic irrational $R'/R$, this shift is achieved by a two-step gauging procedure:

1. First we gauge $U(1)_w$ with flat connections. The gauged theory is the non-compact boson, where all the winding modes are not gauge invariant. Notice that the $U(1)_m$ symmetry gets extended by the $\mathbb{Z}$ quantum symmetry so that we get an $\mathbb{R}$ global symmetry.

2. We gauge a $\mathbb{Z}$ subgroup of the $\mathbb{R}$ symmetry, generated by shifts of period $2\pi R'$. The gauged theory is the compact boson at radius $R'$. The $U(1)_w$ symmetry emerges as the quantum symmetry of the gauged $\mathbb{Z}$ symmetry, while $U(1)_m = \mathbb{R}/\mathbb{Z}$. Notice that, due to the non trivial exact sequence $1 \to \mathbb{Z} \to \mathbb{R} \to U(1) \to 1$, the two $U(1)$ symmetries have a mixed 't Hooft anomaly [53].

Therefore combining 1. and 2. we effectively shift the radius from $R$ to $R'$. Notice that when $R' = \frac{q}{p}R$, then 1. + 2. is an alternative but equivalent route to the $\mathbb{Z}_p \times \mathbb{Z}_q \subset U(1)_m \times U(1)_w$ gauging. In Section 3 we describe this procedure in more detail, at the Lagrangian level.

Let us finally comment on the fact that there is a dual formulation of the bulk theory. This dual formulation comes from interchanging the roles of $b^+$ and $b^-$ in (2.1).[11] All the steps would go through similarly, introducing now an edge mode $\widetilde{\Phi}$. Aiming to trivialize the same lines $U_{\frac{1}{R}n}, V_{Rw}$, the roles of $R$ and $R^{-1}$ must also be interchanged, so that in particular $b^- = -R^{-1}d\widetilde{\Phi}$ on $\overline{\Sigma}_2$. Now enforcing (2.10), namely $b^+ = i \star_2 b^-$ on $\Sigma_2$, and performing the slab compactification, one eventually finds the action for a scalar field $\widetilde{\Phi}$ of radius $1/R$, i.e. the T-dual theory of that in (2.13). Indeed, in this setting the relation (2.10) is equivalent to $d\widetilde{\Phi} = iR^2 \star_2 d\Phi$.

---

[11] The two formulations differ by a boundary term.

# 3 Gauging and Self-Duality Symmetries in the Compact Boson

One of the benefits of working with $\mathbb{R}$ valued gauge fields in the bulk SymTFT is that it allows us to perform topological manipulations sending the radius $R$ to any new $R'$. This is done via the two-step gauging procedure introduced at the end of Section 2.2. This is in contrast to the $U(1)$ BF theory where we can only perform the discrete gaugings $\mathbb{Z}_p \times \mathbb{Z}_q \subset U(1)_m \times U(1)_w$. In this section, we will describe these gauging procedures from the path integral perspective of the 2d theory of a compact boson $\Phi \sim \Phi + 2\pi$, described by the action (2.13) and whose (genuine) local operators are the vertex operators defined as

$$\mathcal{V}_{n,w}(x) =: \exp\left(in\Phi(x)\right)\exp\left(iw\widetilde{\Phi}(x)\right) :, \tag{3.1}$$

where $n, w \in \mathbb{Z}$ and $\widetilde{\Phi}$ is the dual scalar, describing the winding modes of the compact boson.

Let us first start with the more familiar case of gauging $\mathbb{Z}_q \subset U(1)_w$. We perform gauging in the path integral by adding a term to the action coupling a dynamical gauge field $A \in U(1)$ to the conserved current $\star J = \frac{i}{2\pi}d\Phi$:

$$S^{\mathbb{Z}_q \text{ gauged}} = \frac{R^2}{4\pi}\int d\Phi \wedge \star d\Phi + \frac{i}{2\pi}\int d\Phi \wedge A - \frac{iq}{2\pi}\int d\Phi' \wedge A. \tag{3.2}$$

In the third term, the coupling to the scalar $\Phi' \in U(1)$ ensures that $A$ is quantum mechanically equivalent to a $\mathbb{Z}_q$ gauge field. That is to say, summing over the fluxes of $\Phi'$, $\int d\Phi' \in 2\pi\mathbb{Z}$, imposes that

$$\int A \in \frac{2\pi\mathbb{Z}}{q}. \tag{3.3}$$

Performing instead the path integral over $A$ gives the delta function,

$$\delta(d\Phi = q d\Phi') \tag{3.4}$$

such that integrating over $\Phi$ imposes the above relation and we have the new theory

$$S^{\mathbb{Z}_q \text{ gauged}} = \frac{R^2 q^2}{4\pi}\int d\Phi' \wedge \star d\Phi', \tag{3.5}$$

which is the compact boson at the new radius $R' = Rq$. At the level of the local operators, such a gauging tells us that now only the winding vertex operators with charges $w = w'q$ are genuine, and they can be written as $e^{iw'\widetilde{\Phi}'}$. Likewise, it incorporates twisted sectors[12] as the momentum vertex operators with charges $n = n'/q$ become genuine. They are indeed written as $e^{in'\Phi'}$.

---

[12]The introduction of twisted sectors is important to ensure modular invariance of the 2d CFT.

Similarly, we can perform the discrete gauging $\mathbb{Z}_p \subset U(1)_m$ of the shift symmetry. To do so, we add a term to the action with the gauge field $A$ coupled to the conserved current $\star J = \frac{R^2}{2\pi} \star d\Phi$:

$$S^{\mathbb{Z}_p \text{ gauged}} = \frac{R^2}{4\pi} \int (d\Phi - A) \wedge \star(d\Phi - A) - \frac{\mathrm{i}p}{2\pi} \int d\widetilde{\Phi}' \wedge A . \tag{3.6}$$

Again, the path integral over $\widetilde{\Phi}' \in U(1)$ imposes the condition $\int A \in \frac{2\pi\mathbb{Z}}{p}$. The path integral over $A$ gives instead the delta function

$$\delta(R^2 \star (d\Phi - A) = \mathrm{i}pd\widetilde{\Phi}') \tag{3.7}$$

such that eventually

$$S^{\mathbb{Z}_p \text{ gauged}} = \frac{p^2}{4\pi R^2} \int d\widetilde{\Phi}' \wedge \star d\widetilde{\Phi}' . \tag{3.8}$$

Note however that the $\star$ in the delta function (3.7) means that we are dualizing the action also, sending $n \leftrightarrow w$ and inverting the radius along with a rescaling.[13] We see the rescaling describes the compact boson $\Phi'$ at radius $R' = R/p$. The genuine operators in the $\Phi'$ theory are those with charges $(n, w) = (n'p, w'/p)$.

The combined gauging of $\mathbb{Z}_p \times \mathbb{Z}_q \subset U(1)_m \times U(1)_w$ is only possible for $\gcd(p, q) = 1$ due to the mixed t'Hooft anomaly between the two $U(1)$ factors. When the radius is $R^2 = \frac{p}{q}$, gauging the subgroup $\mathbb{Z}_p \times \mathbb{Z}_q \subset U(1)_m \times U(1)_w$ along with T-duality, results in a non-trivial operation that maps the theory onto itself. This is the mechanism for generating the non-invertible T-duality symmetry as described in [42].

Now, we can move on to the more interesting case, describing the two-step gauging procedure to send $R \to R'$. The first step is to gauge the entire $U(1)_w$ with flat connections. We do so by adding a term to the action coupling the conserved current $\star J = \frac{\mathrm{i}}{2\pi} d\Phi$ to the gauge field $A \in U(1)$:

$$S^{U(1)_w \text{ gauged}} = \frac{R^2}{4\pi} \int d\Phi \wedge \star d\Phi + \frac{\mathrm{i}}{2\pi} \int d\Phi \wedge A - \frac{\mathrm{i}}{2\pi R} \int d\phi' \wedge A . \tag{3.9}$$

The third term couples the gauge field to the scalar $\phi' \in \mathbb{R}$ which acts as a Lagrange multiplier enforcing the flatness of $A$. The prefactor is arbitrary, and we have fixed it for future convenience. As $\phi'$ is $\mathbb{R}$-valued, it has no winding $\int d\phi' = 0$, and therefore will not restrict the holonomies of $A$. The integral over $A$ supplies the delta function

$$\delta(d\Phi = R^{-1}d\phi') . \tag{3.10}$$

---

[13]In other words, we see that the above procedure of gauging (a discrete subgroup of) the momentum symmetry in the path integral effectively implements T-duality at the same time. In fact, when $p = 1$ this procedure is exactly performing the T-duality, sending $R \to 1/R$ and $n \leftrightarrow w$. In that case, it is an invertible operation since we are only gauging the identity.

This implies that we are picking out the non-winding sector for the boson, making the resulting theory indistinguishable from the non-compact boson,

$$S^{U(1)_w \text{ gauged}} = \frac{1}{4\pi} \int d\phi' \wedge \star d\phi' \tag{3.11}$$

with $\mathbb{R}$ momentum symmetry. This means that momentum vertex operators are genuine for all $n \in \mathbb{R}$, while no winding vertex operators are genuine for any value of $w \neq 0$.

The second step of the procedure is to gauge a $\mathbb{Z}$ subgroup of the $\mathbb{R}$ momentum symmetry. This is achieved by the following coupling

$$S^{\mathbb{Z} \circ U(1)_w \text{ gauging}} = \frac{1}{4\pi} \int (d\phi' - a) \wedge \star (d\phi' - a) - \frac{i}{2\pi R'} \int d\widetilde{\Phi}'' \wedge a , \tag{3.12}$$

where $a$ is now an $\mathbb{R}$-valued gauge field and in the last term we have coupled $a$ to $\widetilde{\Phi}'' \in U(1)$ such that $\int a \in 2\pi R' \mathbb{Z}$. Integrating over $a$ imposes,

$$\delta(\star(d\phi' - a) = \frac{i}{R'} d\widetilde{\Phi}'') \tag{3.13}$$

such that

$$S^{\mathbb{Z} \circ U(1)_w \text{ gauging}} = \frac{1}{4\pi (R')^2} \int d\widetilde{\Phi}'' \wedge \star d\widetilde{\Phi}'' . \tag{3.14}$$

Note that, as in the discrete case, the inclusion of the Hodge star in the delta function means we are also T-dualizing. The resulting theory is that of the compact boson $\Phi''$ at radius $R' \in \mathbb{R}$.

Equipped with these gauging procedures, we can define a non-invertible self-duality symmetry for the compact boson at any value of the radius. For generic $R \in \mathbb{R}$ this is from the following steps:

1. Gauging $U(1)_w$ with flat connections.

2. Gauging $\mathbb{Z} \subset \mathbb{R}$ for the shift of period $2\pi \frac{1}{R}$.

3. Performing the $T$-duality sending $1/R \to R$.

Note that the last two steps are performed at the same time in the Lagrangian procedure outlined above.

The corresponding duality defect $\mathcal{N}(\gamma)$ can be constructed by performing this sequential gaugings on half space and imposing Dirichlet boundary conditions for the corresponding gauge fields on the interface $\gamma$.[14]

---

[14]See e.g. [54] for a construction of the T-duality defect of the compact boson at the Lagrangian level.

## 3.1 Action on Vertex Operators

To actually check that $\mathcal{N}(\gamma)$ is a non trivial symmetry operator, we should compute its action on the local operators of the theory. A similar procedure to the one described above can be applied to determine how this duality symmetry acts on the vertex operators of the theory.

For simplicity let us consider the insertion of a vertex operator $\mathcal{V}_{n,0}$, so that the action can be written as

$$S = \int \frac{R^2}{4\pi} d\Phi \wedge \star d\Phi - \int in\Phi\delta(x) \,. \tag{3.15}$$

The first step of gauging $U(1)_w$ gives the action

$$S^{U(1)_w \text{ gauged}} = \frac{R^2}{4\pi} \int d\Phi \wedge \star d\Phi + \frac{i}{2\pi} \int d\Phi \wedge A - \frac{i}{2\pi R} \int d\phi' \wedge A - \int in\Phi\delta(x) \,. \tag{3.16}$$

The integral over $A$ gives the action

$$S^{U(1)_w \text{ gauged}} = \frac{1}{4\pi} \int d\phi' \wedge \star d\phi' - \frac{in}{R} \int \phi'\delta(x) \,, \tag{3.17}$$

describing a non-compact boson with the insertion of a vertex operator of charge $n/R$.

The second step of gauging the subgroup $\mathbb{Z}$ of the shift symmetry $\mathbb{R}$ is described by the action

$$S^{\mathbb{Z} \circ U(1)_w \text{ gauging}} = \frac{1}{4\pi} \int (d\phi' - a) \wedge \star(d\phi' - a) - \frac{iR}{2\pi} \int d\Phi'' \wedge a - \frac{in}{R} \int \phi'\delta(x) \,. \tag{3.18}$$

The integral of $a$ now gives the action

$$S^{\mathbb{Z} \circ U(1)_w \text{ gauging}} = \frac{R^2}{4\pi} \int d\Phi'' \wedge \star d\Phi'' - \frac{in}{R^2} \int \widetilde{\Phi}''\delta(x) \,, \tag{3.19}$$

describing the original theory of a compact boson at radius $R$ but with the insertion of a vertex operator $\mathcal{V}_{0,n/R^2}$. Notice that if $n/R^2 \notin \mathbb{Z}$, the action (3.19) is not gauge invariant and the resulting correlator vanishes.

A similar analysis can be done for vertex operators with a non-trivial winding charge. The generic action of the T-duality defect (up to a prefactor that we are not fixing) is then

$$\mathcal{N} \,:\, \mathcal{V}_{n,w} \to \begin{cases} \mathcal{V}_{R^2 w, \frac{n}{R^2}} & \text{if } R^2 w \in \mathbb{Z} \,, \ \frac{n}{R^2} \in \mathbb{Z} \\ \text{non genuine operators} & \text{otherwise.} \end{cases} \tag{3.20}$$

When $R^2 = \frac{p}{q}$ we recover (up to normalization constants) the action of the duality symmetry described in [42], showing that the sequence of gaugings of $U(1)_w$ and $\mathbb{Z}$ are equivalent to the

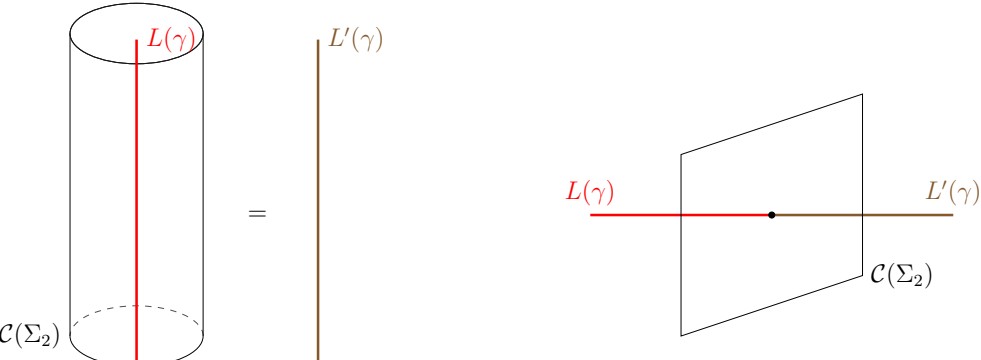

Figure 1: Action of 2d surface operators on line defects. Left: A cylindrical configuration surrounding a line defect. After shrinking the surface, a new line defect is obtained. Right: A junction between the surface and line defects.

discrete gauging of $\mathbb{Z}_p \times \mathbb{Z}_q \subset U(1)_m \times U(1)_w$. On the other hand, when $R^2$ is an irrational number, the T-duality symmetry maps all the genuine vertex operators to non-genuine ones. We notice that this is consistent with the scaling dimension of these operators, given by

$$\Delta_{n,w} = \frac{1}{2}\left(\frac{n^2}{R^2} + w^2 R^2\right).\tag{3.21}$$

Indeed it is trivial to check that

$$\Delta_{n,w} = \Delta_{R^2 w, \frac{n}{R^2}} \qquad \forall R \in \mathbb{R}\,,\tag{3.22}$$

Note that since the two-point functions involving vertex operators are entirely determined by the scaling dimensions $\Delta_{n,w}$, the identity (3.22) imposes selection rules on these functions. In particular, when $R^2 \notin \mathbb{Q}$, these selection rules establish identities between the two-point functions of genuine local operators and those of non-genuine operators connected by a topological line, consistent with the presence of the non-invertible duality symmetry $\mathcal{N}$ (see e.g. [55]). Conversely, when $R^2 \in \mathbb{Q}$, in addition to these relations, we also get identities between correlation functions involving only genuine operators (see also [54] for a recent discussion).

# 4 Condensation Defects and Non-Invertible T-duality

In this section, we show how the SymTFT systematically encodes the duality defects of the boundary theory.

So far, in the three-dimensional BF theory (2.1) we have only considered line operators generating 1-form symmetries. In fact, such a TQFT also enjoys 0-form symmetries, gen-

erated by two-dimensional topological surfaces and acting on line operators as depicted in figure 1. These surface defects are obtained as *condensation defects* [46, 47].

The general construction of condensation defects $C(\Sigma; \mathcal{A})$ is from higher gauging a 1-form symmetry $\mathcal{A}$ on the codimension-1 manifold $\Sigma \subset X_3$. The condensation defects can be expressed as a sum over line insertions along non-trivial cycles:

$$C(\Sigma; \mathcal{A}) = \frac{1}{\#} \sum_{\gamma \in H_1(\Sigma, \mathcal{A})} W(\gamma) , \qquad (4.1)$$

where $W$ is the simplest generating line for the chosen symmetry $\mathcal{A}$. The normalization factor $\frac{1}{\#}$, which in the discrete case is fixed to be $|H_1(\Sigma, \mathcal{A})|^{-1/2}$ [47], will be determined more generally by requiring that acting with the defect on a line returns a line (or sum of lines) with unit prefactor.

From the boundary point of view, these symmetries are generically topological manipulations changing the global variant of the theory. By placing these surface defects on a submanifold of the 3d slab parallel to the boundaries, we can fuse them with the topological boundary, thereby generating a new boundary condition. A characteristic of condensation defects is that they can be opened. As we will see, for specific choices of boundary conditions the open defects induce new 0-form symmetries of the boundary theory, coming from specific self-dualities under some topological manipulation.

Our goal is to define and characterize condensation defects of the $\mathbb{R}$-valued BF theory (see [35] for a previous discussions about condensation defects of a $\mathbb{R}$ global symmetry), thus identifying the aforementioned 0-form symmetries present in theories with $U(1)$ global symmetries.

## 4.1 A Family of Condensation Defects

Our main motivation is to find a condensation defect that reduces to the T-duality defect, i.e. swapping Wilson lines $U \leftrightarrow V$ with the same charges in the bulk, such that it exchanges $R \leftrightarrow \frac{1}{R}$ for any value of the radius.

Let us start by considering condensation defects defined on a submanifold parallel to the boundary, $\Sigma = T^2$, by a higher gauging of an $\mathbb{R}$ subgroup of $\mathbb{R} \times \mathbb{R}$ which we will take to be a general linear combination, generated by the $U$ and $V$ lines living on the same cycle $\gamma$:

$$\begin{aligned} C_s^T(T^2) &\propto \sum_{\gamma \in H_1(\Sigma, \mathbb{R})} U(\gamma) V(-s\gamma) , \\ &\propto \int_{x,y \in \mathbb{R}} U(x\alpha + y\beta) V(-sx\alpha - sy\beta) , \end{aligned} \qquad (4.2)$$

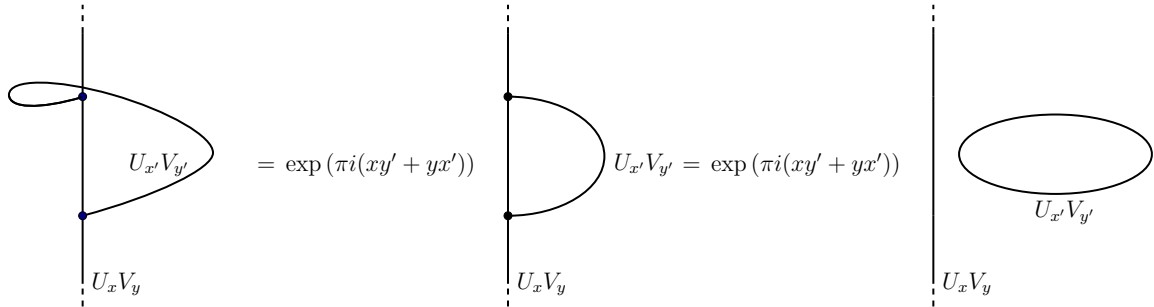

Figure 2: Resolving the junction of lines, to get a normal ordering phase.

where $s \in \mathbb{R}$ is a real coefficient parametrizing a family of condensation defects. In going from the first to the second line of (4.2) we have translated a formal sum into an actual double integral over $\mathbb{R}$ valued charges, and we have parametrized the torus by its 1-cycles $\alpha$, $\beta$.

Introducing a convenient normalization, and a phase after normal ordering $U$ and $V$ on the different cycles (see [9, 47] and figure 2), we get

$$C_s^T(T^2) = s \int_{x,y \in \mathbb{R}} e^{-2\pi i s x y} \, U_y(\beta) V_{-sy}(\beta) U_x(\alpha) V_{-sx}(\alpha) \ . \tag{4.3}$$

We will soon show that the above normalization ensures that $C_s^T$ squares to one (and as a consequence, it is invertible). Note that the normalization precisely excludes a linear combination that is degenerate along one of the $\mathbb{R}$ factors. In fact, the latter defects can be shown to be non-invertible, similarly as discussed in [47] for the discrete case. We will not discuss them further.

The $C_s^T$ defect acts on the line $U_z(\beta)$ as

$$
\begin{aligned}
C_s^T(T^2) U_z(\beta) &= s \int_{x,y} e^{-2\pi i s x y} \, U_y(\beta) V_{-sy}(\beta) U_x(\alpha) V_{-sx}(\alpha) U_z(\beta) \\
&= s \int_{x,y} e^{-2\pi i x s (y+z)} \, U_{y+z}(\beta) V_{-sy}(\beta) \ .
\end{aligned} \tag{4.4}
$$

The integral over $x$ produces a Dirac delta function[15]

$$\int_{-\infty}^{\infty} dx \, e^{2\pi i s x (y+z)} = \frac{1}{s} \delta(y+z) \, , \tag{4.5}$$

---

[15]It is the normalization of the Dirac delta that determines the normalization of $C_s^T$.

such that

$$C_s^T(T^2)U_z(\beta) = \int dy \ \delta(y+z)U_{y+z}(\beta)V_{-sy}(\beta)$$
$$= V_{sz}(\beta) \,. \tag{4.6}$$

Similarly we get

$$C_s^T(T^2)V_t(\beta) = U_{t/s}(\beta) \,. \tag{4.7}$$

Note that the family of condensation defects $C_s^T$ acts effectively on the gauge fields as

$$b_\pm \to s^{\pm 1}b_\mp \,, \tag{4.8}$$

which is a symmetry of the action (2.1).

It is now obvious to check that $C_s^T$ squares to the identity. Indeed

$$(C_s^T(T^2))^2 U_z(\beta) = C_s^T(T^2)V_{sz}(\beta) = U_{\frac{sz}{s}}(\beta) = U_z(\beta) \,, \tag{4.9}$$

and similarly for the action on $V_t(\beta)$. Hence we conclude that $(C_s^T(T^2))^2 = 1$. In Appendix B, we give a different derivation of the same result. Further, we also have that

$$C_s^T(T^2)C_{s'}^T(T^2)U_z(\beta) = C_s^T(T^2)V_{s'z}(\beta) = U_{\frac{s'}{s}z}(\beta) \,,$$
$$C_s^T(T^2)C_{s'}^T(T^2)V_t(\beta) = C_s^T(T^2)U_{t/s'}(\beta) = V_{\frac{s}{s'}t}(\beta) \,, \tag{4.10}$$

so that the fusion of two such defects implements the rescaling

$$b_\pm \to \left(\frac{s'}{s}\right)^{\pm 1} b_\pm \,. \tag{4.11}$$

Note that this is also a symmetry of the action (2.1).[16] Actually, such a fusion of two defects turns out to be a condensation defect obtained by higher-gauging the full $\mathbb{R} \times \mathbb{R}$ symmetry with a torsion phase parametrized by $[\theta] \in H^2(\mathbb{R} \times \mathbb{R}, U(1)) = \mathbb{R}$, namely

$$C_s^T(T^2)C_{s'}^T(T^2) = C^\theta(T^2) = \theta(\theta+1)\int_{a,b,x,y\in\mathbb{R}} e^{-2\pi i(\theta ay - (\theta+1)bx)}U_y(\beta)V_b(\beta)U_x(\alpha)V_a(\alpha) \,. \tag{4.12}$$

with

$$\theta = \frac{1}{\frac{s'}{s}-1} \,. \tag{4.13}$$

This defect has straightforwardly the inverse $C^{\theta^{-1}}(T^2)$, with $\theta^{-1} = 1/(\frac{s}{s'}-1) = -(\theta+1)$.

---

[16]When the action is supplemented with the physical boundary term (2.11), the above rescaling is no longer a symmetry (except for $s = \pm 1$). This will play a major role below.

For $s'/s = -1$, i.e. $\theta = -1/2$, this is a condensation defect that implements charge conjugation in the bulk theory. The construction excludes the case when $s = s'$, since in this case we get $C_s^T C_s^T = \mathbb{1}$.

The invertible 0-form symmetry generated by $C_s^T$ and $C^\theta$ is $O(1, 1; \mathbb{R})$. Interestingly, these surface defects span the circle branch of the $c = 1$ conformal manifold, i.e. the subset of the $c = 1$ conformal manifold with a $U(1) \times U(1)$ global symmetry.[17] This follows from the fact that the entire conformal manifold is generated by topological manipulations which include gauging of continuous symmetries with flat connections. It would be interesting to further explore this correspondence in more complicated situations.

## 4.2 Self-Duality Defect

Let us understand more in detail the action of the above condensation defects on the theory defined on a slab. The boundary conditions that we impose are the conformal ones (2.10) at the physical boundary, and the ones that fix the radius $R$ (2.8) at the topological boundary. Note first that the physical boundary conditions are only preserved by $C_{s=1}^T$, $C^{\theta=-1/2}$, and $C_{s=-1}^T$. These correspond to T-duality, charge conjugation, and their composition, respectively. Hence only (the boundaries of) the latter can aspire at being symmetry defects of a given theory.

Let us focus on $C_{s=1}^T$. It exchanges $U_z$ and $V_z$ lines.[18] Thus in particular it sends the lines $U_{n/R}$ to the lines $V_{n/R} = V_{\widetilde{R}\widetilde{w}}$, and the lines $V_{Rw}$ to $U_{Rw} = U_{\widetilde{n}/\widetilde{R}}$. This can be interpreted as the mapping between two theories defined by:

$$\widetilde{n} = w , \qquad \widetilde{w} = n , \qquad \widetilde{R} = \frac{1}{R} . \tag{4.14}$$

This is the usual T-duality. More precisely, on the physical theory after slab compactification this defect is exchanging momentum and winding modes, and rescaling the radius from $R$ to $1/R$. Since the latter manipulation involves a (discrete) gauging, we expect the symmetry generated by this defect to be non-invertible, except for $R = 1$ where we have a self-T-dual theory and this transformation is an invertible symmetry.

As shown in [8,9] the non-invertible T-duality defect in the boundary theory corresponds, within the SymTFT, to the boundary of an open condensation defect. This defect is defined

---

[17]Notice that if we also include the topological manipulations involving $\mathbb{Z}_2^C$ charge conjugation, we actually produce the entire connected conformal manifold of $c = 1$ CFTs.

[18]As for $C^{\theta=-\frac{1}{2}}$, it implements $n \to -n$, $w \to -w$, i.e. charge conjugation.

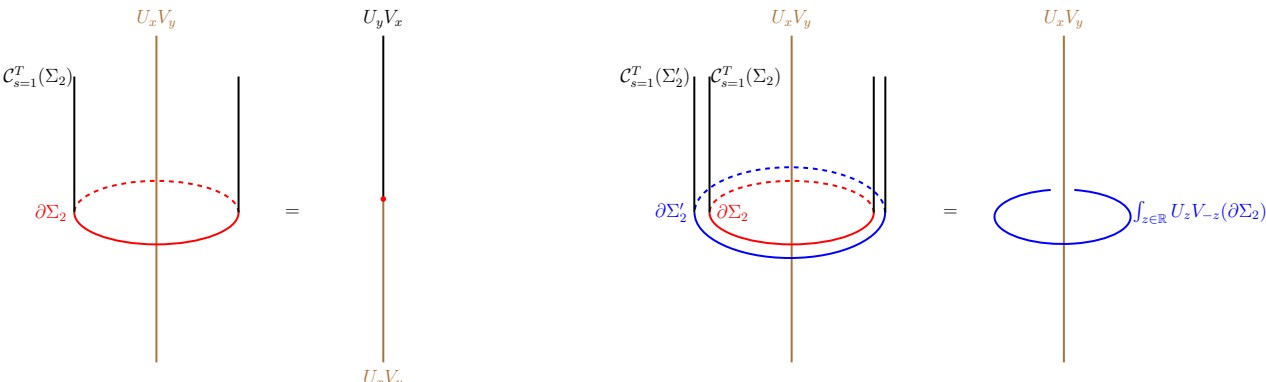

Figure 3: Action of the open condensation defect on a bulk topological line. Left: The action produces a topological junction between two topological line operators. For simple lines this junction reduces to a delta function as explained in equation (4.17). Right: The fusion of two open condensation defects results in a non-simple line on the boundary of the 2-dimensional surface $\Sigma_2$.

by a half-higher gauging with Dirichlet boundary conditions.[19] An easy way to observe the non-invertibility of an open condensation defect $C^T_{s=1}(\mathcal{C})$ is by examining its self-fusion when it has a boundary. The result is a 1-dimensional condensation defect of the same $\mathbb{R}$ subgroup of the bulk $\mathbb{R} \times \mathbb{R}$ symmetry. This arises because the Dirichlet boundary conditions imposed on the boundary of the condensation defect allow the twist defects to absorb the lines that generate the $\mathbb{R}$ subgroup, i.e.

$$\mathcal{C}^T_{s=1}(\Sigma_2) \times U_x V_{-x}(\partial\Sigma_2) = \mathcal{C}^T_{s=1}(\Sigma_2). \tag{4.15}$$

When fusing two condensation defects $C^T_{s=1}$ with boundaries, the resulting object is a genuine line operator, which must be consistent with the property outlined in (4.15), meaning it must act as a projector. Consequently, we obtain:[20]

---

[19]To produce a genuine bulk symmetry operator for a boundary duality defect, we should gauge the $\mathbb{Z}_2$ 0-form symmetry generated by the condensation defect $C^T_s$, with the option of adding discrete torsion $\epsilon \in H^3(\mathbb{Z}_2, U(1))$ which correspond to the Frobenius-Schur indicator of the boundary symmetry [16,17]. It is important to note that the choice of $s$ corresponds to selecting a bicharacter for the duality symmetry [16,17]. Upon gauging, the boundary of this defect becomes a genuine topological line operator with non-invertible fusion properties, and its boundary value reproduces the duality symmetry. This gauged TFT thus serves as the symmetry TFT for the boundary duality defect, together with the invertible symmetries of the theory (see, for example, [8,9]). It describes generalizations of the Tambara-Yamagami fusion category with an infinite number of objects. However, for the purposes of this work, it is sufficient to consider the ungauged theory, which includes all its condensation defects and their associated twist defects.

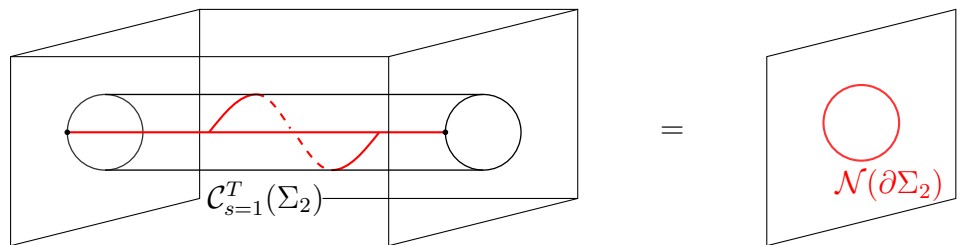

Figure 4: Bulk realization of the boundary duality defect. Left: A condensation defect is placed perpendicularly to the boundary. Right: After the slab compactification, the boundary on the condensation defect generates a topological line operator of the theory.

$$\mathcal{C}_{s=1}^{T}(\Sigma_2) \times \mathcal{C}_{s=1}^{T}(\Sigma_2) \propto \int_{x \in \mathbb{R}} U_x V_{-x}(\partial \Sigma_2) \,. \tag{4.16}$$

This fusion rule is explicitly verified through a detailed computation in Appendix B.1. Additionally, it is worth noting that equation (4.16) aligns well with the action of the open cylinder on a topological line, as illustrated in Figure 3. Specifically, when an open condensation defect acts on a generic bulk line $U_x V_y$, it creates a topological junction between this line and the one obtained by the action of $\mathcal{C}_{s=1}^{T}$. Since $U_x V_y$ (for any $x, y$) are simple lines, the only possible topological junctions are identity operators. Therefore, the open condensation defect projects out the non-invariant lines, resulting in

$$\mathcal{C}_{s=1}^{T}(\Sigma_2) U_x V_y(\gamma) \propto \delta(x - y) U_y V_x(\gamma) \,, \tag{4.17}$$

where $\Sigma_2$ and $\gamma$ arranged as in the left of figure 3. Then we get

$$\begin{aligned}\mathcal{C}_{s=1}^{T}(\Sigma_2) \times \mathcal{C}_{s=1}^{T}(\Sigma_2) U_x V_y(\gamma) &\propto \int_{z \in \mathbb{R}} U_z V_{-z}(\partial \Sigma_2) U_x V_y(\gamma) \\ &\propto \int_{z \in \mathbb{R}} e^{-2\pi i z(x-y)} U_x V_y(\gamma) \propto \delta(x - y) U_x V_y(\gamma) \,,\end{aligned} \tag{4.18}$$

consistently with (4.17).

**Duality defect on the boundary.** To produce the boundary T-duality defect, we must open the condensation defect along one of its cycles and allow it to terminate on the boundaries of the slab. Upon compactification of the slab, this configuration gives rise to a codimension-1 topological operator in the boundary theory, which we denote by $\mathcal{N}(\partial \Sigma_2)$ (see figure 4).

---

[20]In the following discussion, we will frequently omit the proportionality factors involved in the fusion process. These factors are computed in Appendix B and will be briefly discussed at the end of this section.

We would like to show how the boundary properties of the duality symmetry emerge from the bulk perspective. To do so, consider the open condensation defect as a cylinder $\mathcal{C}$, with the $\alpha$ cycle parallel to the boundary and the $\beta$ cycle being the one with a boundary. Only the endable lines, determined by our choice of topological boundary conditions, will live along the $\beta$ direction, and the symmetry-generating lines along the $\alpha$ direction. The sum over $U, V$ along the same cycle is now only sensible when the charge lattices of the $U$ and $V$ lines are overlapping.

We choose topological boundary conditions as in (2.8), ensuring that both $U_{\frac{n}{R}}(\beta)$ and $V_{Rw}(\beta)$ with $n, w \in \mathbb{Z}$ are trivialized at the boundary. Consequently, the integral over $y$ of $U_y(\beta)$ simplifies to a sum over $n$, yielding $U_{y=n/R}(\beta)$. For $s = 1$, we must determine which of the $V_{-y=-n/R}(\beta)$ are also trivializable. These correspond to values of $n$ for which there exists an integer $w$ such that $-n/R = wR$. Therefore, for irrational $R^2$, the overlap reduces to the identity operator so that only the integral over the $\alpha$ direction remains non-trivial. On the contrary, for $R^2 = p/q$ the integral over the $\beta$ direction reduces to a discrete sum, so that the condensation defect can be written as

$$\mathcal{C}_{s=1}^T(\Sigma_2; \partial\Sigma_2 \subset \partial X_3) = \sum_{k \in \mathbb{Z}} \int_{x \in \mathbb{R}} e^{-2\pi i \sqrt{pq} kx} U_{\sqrt{pq}k} V_{-\sqrt{pq}k}(\beta) U_x V_{-x}(\alpha) \quad \text{if } R^2 = \frac{p}{q}. \quad (4.19)$$

In this configuration, $\mathcal{C}_{s=1}^T$ will act by projecting out all the endable lines that are not in the overlapping sub-lattice, hence in particular all the non-trivial lines in the irrational case.[21] It is then clearly non-invertible, except when $R = 1$. Generically we get

$$\mathcal{C}_{s=1}^T(\Sigma_2) U_{\frac{n}{R}} V_{Rw}(\gamma) \propto \begin{cases} U_{\frac{R^2 w}{R}} V_{R(\frac{n}{R^2})} & \text{if } R^2 w \in \mathbb{Z}, \ \frac{n}{R^2} \in \mathbb{Z} \\ \text{non genuine operators} & \text{otherwise} \end{cases} \quad (4.20)$$

where $\partial\Sigma_2 \subset \partial X_3, \partial\gamma \subset \partial X_3$ and $\text{Link}(\partial\gamma, \partial\Sigma_2) = 1$. After the slab compactification, this implies the following action on the primary operators of the boundary theory

$$\mathcal{N}(\partial\Sigma_2) \mathcal{V}_{n,w} \propto \begin{cases} \mathcal{V}_{R^2 w, \frac{n}{R^2}} & \text{if } R^2 w \in \mathbb{Z}, \ \frac{n}{R^2} \in \mathbb{Z} \\ \text{non genuine operators} & \text{otherwise.} \end{cases} \quad (4.21)$$

This nicely reproduces the action on the vertex operators described in section 3.1.

Finally, let us compute the fusion of two duality defects. This can be achieved by placing two concentric defects, each ending perpendicularly on the two boundaries. As described earlier, the integral over the lines in the $\beta$ direction, perpendicular to the boundary, depends

---

[21]Note that for rational $R^2$, this projection is less severe than in the case of an open defect in the bulk (4.17). As we will argue, this is closely related to the fact that, at these specific radii, the boundary theory is self-dual under discrete (rather than continuous) gauging.

on the topological boundary conditions. When $R^2$ is irrational, the integral reduces to the identity, and the fusion of two condensation defects becomes

$$\mathcal{C}^T_{s=1}(\Sigma_2) \times \mathcal{C}^T_{s=1}(\Sigma_2) \propto \int_{x \in \mathbb{R}} U_x V_{-x}(\partial\Sigma_2) \,, \tag{4.22}$$

as in the bulk analysis.

After the slab compactification, we can impose the physical boundary conditions on this expression, yielding

$$\mathcal{N}(\partial\Sigma_2) \times \mathcal{N}(\partial\Sigma_2) \propto \int_{x \in \mathbb{R}} U_x V_{-x}(\partial\Sigma_2) \equiv \int_{x \in \mathbb{R}} U^{(w)}_{-2\pi x R} U^{(m)}_{2\pi \frac{x}{R}} \,, \tag{4.23}$$

where $U^{(m)}_a U^{(w)}_b$ (with $a, b \in [0, 2\pi)$) are the topological operators generating the $U(1)_m \times U(1)_w$ symmetry of the boundary compact boson. Notice that the combination $U^{(w)}_{-2\pi x R} U^{(m)}_{2\pi \frac{x}{R}}$ (with $x \in \mathbb{R}$) generates a non-compact direction of the $U(1) \times U(1)$ global symmetry.

In contrast, when $R^2 = p/q$, the integral over the $\beta$ direction reduces to an infinite sum. Thus, the fusion of two condensation defects in this case becomes

$$\mathcal{C}^T_{s=1}(\Sigma_2) \times \mathcal{C}^T_{s=1}(\Sigma_2) \propto \sum_{k,k' \in \mathbb{Z}} \int_{x,y \in \mathbb{R}} e^{-2\pi i \sqrt{pq}(kx+k'y+2k'x)} U_{\sqrt{pq}(k+k')} V_{-\sqrt{pq}(k+k')}(\beta) U_{x+y} V_{-(x+y)}(\alpha)$$

$$= \sum_{k \in \mathbb{Z}} U_{k/\sqrt{pq}} V_{-k/\sqrt{pq}}(\alpha) \,. \tag{4.24}$$

After the slab compactification, we obtain

$$\mathcal{N}(\partial\Sigma_2) \times \mathcal{N}(\partial\Sigma_2) \propto \sum_{k \in \mathbb{Z}} U_{k/\sqrt{pq}} V_{-k/\sqrt{pq}}(\partial\Sigma_2) = \sum_{k \in \mathbb{Z}} U^{(w)}_{-\frac{2\pi}{q}k} U^{(m)}_{\frac{2\pi}{p}k}(\partial\Sigma_2) \,, \tag{4.25}$$

which, up to an infinite proportionality factor, is the sum of lines generating the $\mathbb{Z}_p \times \mathbb{Z}_q \subset U(1)_m \times U(1)_w$ symmetry of the boundary theory. We have thus shown that the bulk condensation defect reproduces all possible duality defects of the boundary theory.

We now briefly discuss the normalization of the self-duality defect $\mathcal{N}$. In both equations (4.25) and (4.23), an infinite proportionality factor appears, which can be absorbed into the definition of the boundary duality operator $\mathcal{N}$. However, the normalization of a symmetry defect is constrained by locality conditions. Specifically, for $R^2 = \frac{p}{q}$, the normalization of $\mathcal{N}$ can be established using modular bootstrap arguments, starting with the torus partition function of the compact boson. Inserting the defect along the spatial cycle projects onto states invariant under $\mathcal{N}$. After performing an $S$ transformation, for $\mathcal{N}$ to be a well-defined topological defect it must produce a well-defined twisted Hilbert space, which dictates the correct normalization (see, e.g., [42, 44]). To achieve the appropriate normalization for $R^2 =$

$\frac{p}{q}$, the open condensation defect should be properly normalized by dividing by the infinite factor obtained in (4.24), ensuring that the final result is free from infinities. When $R^2 \notin \mathbb{Q}$ the non-invertible defect involves the gauging of a continuous symmetry and we expect the twisted Hilbert space to contain a continuum of states. Consequently, the analysis applicable to the discrete case does not straightforwardly extend to this scenario. Although a detailed exploration is beyond the scope of this work, it would be interesting to further investigate the Hilbert space interpretation of these symmetries and to rigorously define their normalizations.

# 5 Analogies and Differences with a $U(1)$ BF Symmetry TFT

To emphasize the conceptual difference with the standard case of $U(1)$-valued BF theory, we will briefly summarize how to construct boundary conditions and the condensation defects for these theories (see [47] for a previous discussion). The bulk theory is described by the action

$$S_{U(1)} = \frac{\mathrm{i}N}{2\pi} \int_{X_3} B^+ dB^- \, , \tag{5.1}$$

with $N$ an integer that cannot be rescaled away because of the quantization of the fluxes of $B^\pm$. Line operators of this theory are of the form

$$U_m(\gamma) = \exp\left(\mathrm{i}m \int_\gamma B^+\right) \quad , \quad V_n(\gamma) = \exp\left(\mathrm{i}n \int_\gamma B^-\right) , \tag{5.2}$$

with $n, m = 0, \cdots, N-1$ and braiding $\langle U_m(\gamma) V_n(\gamma') \rangle = \exp\left(\frac{2\pi \mathrm{i}mn}{N} \mathrm{Link}(\gamma, \gamma')\right)$. They generate a $\mathbb{Z}_N \times \mathbb{Z}_N$ 1-form symmetry in the 3d bulk. Under the gauge transformation

$$\delta B^\pm = d\lambda^\pm \qquad \text{with} \qquad \int_\gamma d\lambda^\pm \in 2\pi\mathbb{Z}$$

the action picks up a boundary term

$$\delta S_{U(1)} = \frac{\mathrm{i}N}{2\pi} \int_{\overline{\Sigma}_2} \lambda^+ dB^- \, , \tag{5.3}$$

where we now focus on the topological boundary of the slab. To ensure gauge invariance we couple the theory to a compact scalar edge mode $\Phi \sim \Phi + 2\pi$ with the gauge transformation

$$\Phi \to \Phi - q\lambda^+ \, , \tag{5.4}$$

and with a bulk/boundary action

$$S_{3d/2d} = S_{U(1)} + \frac{\mathrm{i}p}{2\pi} \int_{\overline{\Sigma}_2} \Phi dB^- \, , \tag{5.5}$$

with $q, p \in \mathbb{Z}$. Gauge invariance implies $N = pq$. The boundary equations of motion are

$$B^+|_{\overline{\Sigma}_2} = -\frac{1}{q} d\Phi \ , \tag{5.6}$$

which imply

$$\int_{\gamma \in H_1(\overline{\Sigma}_2, \mathbb{Z})} B^+ \in \frac{2\pi\mathbb{Z}}{q} \ . \tag{5.7}$$

Moreover, the sum over the edge modes' fluxes $\int d\Phi \in 2\pi\mathbb{Z}$ implies

$$\int_{\gamma \in H_1(\overline{\Sigma}_2, \mathbb{Z})} B^- \in \frac{2\pi\mathbb{Z}}{p} \ . \tag{5.8}$$

As a consequence, the lines $U_{qm'}, V_{pn'}$ with $m'(n') = 0, \cdots, p-1(q-1)$ are trivial when pushed parallel to the boundary. More specifically, they can end topologically on the boundary since they are gauge invariant. For $U_{qm'}$, this is seen by defining the *dressed* line operators

$$U_{qm'} = \exp\left( iqm' \int_\gamma B^+ + m' \left. \Phi \right|_{\partial\gamma} \right). \tag{5.9}$$

For $V_{pn'}$, one can provide a similar dressing by a scalar which is the (non-local) boundary dual to $\Phi$, as we will comment below. The non-trivial boundary topological operators generate a $\mathbb{Z}_q \times \mathbb{Z}_p$ 0-form global symmetry on the boundary. Therefore, the full set of topological boundary conditions is classified by the integers $q$ (or $p$) which are divisors of $N$. Notice that the same classification arises by constructing Lagrangian algebras $\mathcal{L}_q = (U_{qm'}, V_{\frac{N}{q}n'})$ corresponding to a maximal set of commuting lines. They exactly correspond to the ones trivialized by the boundary condition. The extreme cases are $q = 1$ (Dirichlet for $B^+$) and $p = 1$ (Dirichlet for $B^-$), for which the 0-form symmetry on the boundary corresponds to either $\mathbb{Z}_N$ of the bulk $\mathbb{Z}_N \times \mathbb{Z}_N$ 1-form symmetry.

## 5.1 Physical Boundary Conditions and Rational Radius

We would now like to impose the conformal boundary conditions at the physical boundary of the slab, similarly to (2.10). Before proceeding, let us underscore what physical system we are aiming to describe: it is a theory of a compact boson, for which we are singling out a $\mathbb{Z}_N$ subgroup of its $U(1) \times U(1)$ global 0-form symmetry. For Dirichlet topological boundary conditions for either $B^+$ or $B^-$, the $\mathbb{Z}_N$ is embedded in either $U(1)$. We can assume without loss of generality that we start from such a situation. Then, the only topological manipulations that are allowed in this setup correspond to gauging all or part of this $\mathbb{Z}_N$. Hence the different theories are in one-to-one correspondence with divisors of $N$.

Note that now we cannot rescale freely the discrete gauge fields $B^\pm$. We will see later that some peculiar rescalings can be implemented, however for the moment let us assume

that integrality of their fluxes fixes the normalization. As a consequence, the coefficient of the analog of (2.11) is now physical. We thus write:

$$S_{2d/3d} = S + \frac{1}{4\pi R_0^2} \int_{\Sigma_2} B^- \wedge \star_2 B^- . \tag{5.10}$$

The boundary conditions are then

$$B^-|_{\Sigma_2} = \mathrm{i} N R_0^2 \star_2 B^+|_{\Sigma_2} . \tag{5.11}$$

We then proceed exactly as in the continuous case with the slab compactification, first getting $B^- = -\mathrm{i} p R_0^2 \star_2 d\Phi$ and then finally

$$S_{2d} = \frac{p^2 R_0^2}{4\pi} \int_{\Sigma_2} d\Phi \wedge \star_2 d\Phi . \tag{5.12}$$

This means that the physical radius is $R = p R_0$. In particular, in going from Dirichlet boundary conditions for $B^+$ ($p = N$) to Dirichlet for $B^-$ ($p = 1$), one goes from radius $R = N R_0$ to radius $R' = R_0 = R/N$. When Dirichlet is on $B^+$, the boundary $\mathbb{Z}_N$ symmetry is the momentum one. Hence we recover that fully gauging this momentum symmetry gives back a theory where the radius is $N$ times smaller. If $N$ has divisors, then by gauging $\mathbb{Z}_q \subset \mathbb{Z}_N$ we can go from $R = N R_0$ to $R' = p R_0 = R/q$. We conclude that the $U(1)$ BF theory allows us to describe the compact boson at any radius, but this comes at the cost of coupling the theory to backgrounds for only a $\mathbb{Z}_N \subset U(1)_m \times U(1)_w$ symmetry, thus with the possibility of performing only its corresponding topological manipulations. In this sense, the $U(1)$ BF theory provides less information about the boundary compact boson compared to the non-compact $\mathbb{R}$-valued BF theory discussed in the previous sections.

In analogy to the continuous case, the bulk theory has a dual formulation where we exchange $B^+$ and $B^-$ and ensure that the same lines are trivialised on the boundary. We find that this dual formulation in the bulk implements the T-duality upon slab compactification. In detail, this works as follows. Switching the roles of $B^+$ and $B^-$, we see that the same conformal boundary conditions (5.11) are obtained by writing the analog of (5.10) with $B^+$ and replacing $R_0$ by $\widetilde{R}_0 = (N R_0)^{-1}$. In order to get the same boundary ending lines in (5.7) and (5.8), the topological boundary condition corresponds to that imposing $B^-|_{\bar{\Sigma}_2} = -\frac{1}{p} d\widetilde{\phi}$. Compactifying the interval we get

$$S_{2d} = \frac{1}{4\pi p^2 R_0^2} \int_{\Sigma_2} d\widetilde{\Phi} \wedge \star_2 d\widetilde{\Phi} \tag{5.13}$$

which is exactly the T-dual of (5.12). This is a completely equivalent description of the same theory. Hence we can contemplate performing this T-duality to undo the rescaling of the radius caused by a gauging. More precisely, given a rational radius $R^2 = \frac{p}{q}$, we

can select a symmetry $\mathbb{Z}_p^n \times \mathbb{Z}_q^w \subset U(1)_m \times U(1)_w$ such upon gauging, the radius becomes $R' = \frac{q}{p}\sqrt{\frac{p}{q}}$. Then performing the $T$-duality sends us back to the original radius. Again, this is a non-invertible operation as it involves discrete gaugings (unless we are at the self-dual radius).

Let us finally comment on the possibility to perform an effective (discrete) rescaling of $B^{\pm}$. In principle, for any $N$, one can take an $l$ such that $\gcd(N, l) = 1$ and generate all the lines with $U_l$ and $V_{(l^{-1})_N}$, where $(l^{-1})_N$ is an integer in $\mathbb{Z}_N$ such that $(l^{-1})_N l = 1 \mod N$. Note that also the braiding of the lines is preserved. At this stage it is not clear what this operation does on the physical theory, because clearly we cannot apply the rescaling to the action (5.1) without spoiling the normalization (which is fixed for us by the fact that we focus on the $\mathbb{Z}_N$ symmetry of the physical theory). We will see below what condensation defect implements such a rescaling, and how it can affect the topological and/or physical boundary conditions.

## 5.2    Condensation Defects for the $U(1)$ BF Theory

As in the continuous case, we can define a family of (invertible) condensation defects from higher gauging on $\Sigma = T^2$ a $\mathbb{Z}_N$ subgroup of $\mathbb{Z}_N \times \mathbb{Z}_N$. They are labeled this time by an integer $k$, which must have $\gcd(N, k) = 1$ and in particular, is non-vanishing.[22] Using a natural normalization for finite groups, we can write

$$
\begin{aligned}
C_k^{\mathbb{Z}_N}(T^2) &= |H_1(T^2, \mathbb{Z}_N)|^{-1/2} \sum_{\gamma \in H_1(T^2, \mathbb{Z}_N)} U(\gamma)V(-k\gamma) \\
&= \frac{1}{N} \sum_{a,b=0}^{N-1} U(a\alpha + b\beta)V(-ka\alpha - kb\beta) .
\end{aligned}
\tag{5.14}
$$

We have parametrized the torus by its 1-cycles $\alpha$, $\beta$, with charges $a, b \in \mathbb{Z}_N$ along the respective cycles. There is a need for a normal ordering phase $e^{-2\pi i \frac{kab}{N}}$ to account for the ordering ambiguities of $U$ and $V$. We then have

$$
C_k^{\mathbb{Z}_N}(T^2) = \frac{1}{N} \sum_{a,b=0}^{N-1} e^{-2\pi i \frac{kab}{N}} U_b(\beta)V_{-kb}(\beta)U_a(\alpha)V_{-ka}(\alpha) .
\tag{5.15}
$$

---

[22]When $\gcd(N, k) \neq 1$ or $k = 0$ the resulting defect is non-invertible.

Let us consider the defect acting on the line operator $U_n(\beta)$:

$$C_k^{\mathbb{Z}_N}(T^2)U_n(\beta) = \frac{1}{N}\sum_{a,b=0}^{N-1}e^{-2\pi i\frac{kab}{N}}U_b(\beta)V_{-kb}(\beta)U_a(\alpha)V_{-ka}(\alpha)U_n(\beta)$$

$$= \frac{1}{N}\sum_{a,b=0}^{N-1}e^{-2\pi i\frac{a}{N}k(b+n)}U_{b+n}(\beta)V_{-kb}(\beta)$$

(5.16)

The sum over $a$ gives the Kronecker delta function $N\delta(kb+kn=0 \bmod N)$ which, since we assume $\gcd(N,k)=1$, is equivalent to $N\delta(b+n=0 \bmod N)$. We then have

$$C_k^{\mathbb{Z}_N}(T^2)U_n(\beta) = V_{kn}(\beta)\,.$$

(5.17)

The action on $V_w(\beta)$ yields

$$C_k^{\mathbb{Z}_N}(T^2)V_w(\beta) = U_{(k^{-1})_N w}(\beta)\,,$$

(5.18)

where as before, $k(k^{-1})_N = 1 \bmod N$. Note that for $k = 1$, this condensation defect just swaps the $U$ and $V$ lines. For $k > 1$, the parameter $k$ serves to reshuffle the Wilson lines. That is, it maps a generator of $\mathbb{Z}_N$ to another generator of $\mathbb{Z}_N$, such that it is an automorphism of $\mathbb{Z}_N$. From the way it acts on Wilson lines, the defect $C_k^{\mathbb{Z}_N}$ appears to effectively map the gauge fields $B^+ \to kB^-$ and $B^- \to (k^{-1})_N B^+$, swapping gauge fields and also performing the rescaling described at the end of the previous section.

As in the continuous case, the above transformations straightforwardly show that $(C_k^{\mathbb{Z}_N})^2 = \mathbb{I}$. We can also compose two defects with different labels. Their action is

$$C_k^{\mathbb{Z}_N}(T^2)C_{k'}^{\mathbb{Z}_N}(T^2)U_n(\beta) = C_k^{\mathbb{Z}_N}(T^2)V_{k'n}(\beta) = U_{(k^{-1})_N k'n}\,,$$
$$C_k^{\mathbb{Z}_N}(T^2)C_{k'}^{\mathbb{Z}_N}(T^2)V_w(\beta) = C_k^{\mathbb{Z}_N}(T^2)U_{(k'^{-1})_N w}(\beta) = V_{k(k'^{-1})_N w}\,.$$

(5.19)

This is a combined rescaling as at the end of the previous subsection, with $l = (k^{-1})_N k'$ (which also satisfies $\gcd(N,l) = 1$ since both $k$ and $k'$, and hence their inverses mod $N$, do). For $k = 1$ and $k' = -1$, the combination of condensation defects implements charge conjugation. In all generality, we can see the combination of two defects as a higher gauging of the full $\mathbb{Z}_N \times \mathbb{Z}_N$, with torsion.

We can also consider condensation defects constructed by higher gauging a subgroup $\mathbb{Z}_q \subset \mathbb{Z}_N \times \mathbb{Z}_N$, for $q$ a divisor of $N$. We define them as

$$C_k^{\mathbb{Z}_q}(T^2) = \frac{1}{q}\sum_{a,b=0}^{q-1}e^{-2\pi i\frac{pkab}{q}}U_{pb}(\beta)V_{-kpb}(\beta)U_{pa}(\alpha)V_{-kpa}(\alpha)$$

(5.20)

where $p = N/q$ and $\gcd(k, q) = 1$. Its action on the line defects of the theory is

$$C_k^{\mathbb{Z}_q}(T^2)U_n(\beta) = \sum_{b \in \mathbb{Z}_q} \delta(pb + n = 0 \mod q)U_{pb+n}(\beta)V_{-pkb}(\beta)$$

$$C_k^{\mathbb{Z}_q}(T^2)V_m(\beta) = \sum_{b \in \mathbb{Z}_q} \delta(m - pkb = 0 \mod q)U_{pb}(\beta)V_{-pkb+m}(\beta).$$

(5.21)

The action of the defect is invertible when $\gcd(p, q) = 1$, which we verify through fusion arguments provided in Appendix B.2.

**Non-Invertible Defects.** As in the continuous case, we wish to interpret how these defects act on the theory defined on the slab. Thus, we wish to pick out which of the (boundaries of) defects will act as symmetry defects for the 2d theory.

Firstly, we see that the BF action (5.1) is only invariant under swapping $B^+ \leftrightarrow \pm B^-$ which is implemented by the $C_k^{\mathbb{Z}_N}$ defect for $k = \pm 1$. Secondly, the physical boundary conditions (5.11) are only conserved by $C_{k=\pm 1}^{\mathbb{Z}_N}$ in the case when $R_0^2 = N^{-1}$. This means that when we compactify the interval, with suitable physical boundary conditions such that the 2d theory is at radius $R = \sqrt{\frac{p}{q}}$, the defects $C_{k=1}^{\mathbb{Z}_N}$, the fused defect $C_{k=1}^{\mathbb{Z}_N} \times C_{k=-1}^{\mathbb{Z}_N}$, and $C_{k=-1}^{\mathbb{Z}_N}$ will produce T-duality, charge conjugation, and their composition, respectively. The limitations in the discrete case of defining T-duality and charge conjugation defects again highlight the usefulness of the non-compact BF theory, where we were able to define such defects for any value of the radius.

In the case that $R_0 = \frac{1}{\sqrt{N}} = \frac{1}{\sqrt{pq}}$ and $R = \sqrt{\frac{p}{q}}$, the boundaries of these defects, once we compactify the interval, will become (possibly non-invertible) symmetry generators in the 2d theory. We can define such non-invertible defects by cutting open a hole along one of the cycles of the defect and pulling it to the topological boundary. To demonstrate this, we can construct the defect on the cylinder $\mathcal{C}$, with the $\alpha$ cycle parallel to the boundary and the $\beta$ cycle ending on the boundary. Now, based on the chosen boundary conditions, only lines that can end trivially on the topological boundary are along the $\beta$ direction, and the remaining symmetry-generating lines are along the $\alpha$ direction. The sum over $U, V$ along the same cycle in the defect, now is only sensible when summing over the overlapping sub-lattices of charges. This will also alter the normalization of the defect.

Let us show this construction explicitly for the $C_{k=1}^{\mathbb{Z}_N}$ defect. We take the topological boundary conditions such that the lines $U_{qm}(\beta)$ and $V_{p\ell}(\beta)$ with $m \in \mathbb{Z}_p$ and $\ell \in \mathbb{Z}_q$ end trivially on the boundary. The two lattices of charges of boundary ending lines only overlap for the charges $\in lcm(p, q)\mathbb{Z}_{gcd(p,q)}$. The remaining lines along the $\alpha$ direction are $U_r(\alpha)$ and $U_s(\alpha)$ with $r \in \mathbb{Z}_p$ and $s \in \mathbb{Z}_q$. Asking that $p > q$, these lattices of charges overlap on $\mathbb{Z}_q$.

The $C^{\mathbb{Z}_N}_{k=1}(\Sigma_2)$ defect perpendicular to the boundary can be written as

$$C^{\mathbb{Z}_N}_{k=1}(\Sigma_2) = \frac{1}{q}\sum_{r-0}^{q-1}\sum_{\ell=0}^{gcd(p,q)-1} e^{-2\pi i\frac{\ell r}{gcd(p,q)}}\, U_{lcm(p,q)\ell}(\beta)V_{-lcm(p,q)\ell}(\beta)U_r(\alpha)V_{-r}(\alpha)\ . \tag{5.22}$$

Analogously to the continuous case, the defect will swap $U \leftrightarrow V$ while also projecting out all endable lines that do not lie within the overlapping sub-lattice. Its action on lines is summarized by

$$C^{\mathbb{Z}_N}_{k=1}(\Sigma_2)U_{qn}V_{pw}(\gamma) = \begin{cases} V_{qn}U_{pw}(\gamma) & \text{if } n \in \frac{lcm(p,q)}{q}\mathbb{Z},\ w \in \frac{lcm(p,q)}{p}\mathbb{Z} \\ \text{non-genuine operators} & \text{otherwise} \end{cases} \tag{5.23}$$

where $\partial\Sigma_2 \in \partial X_3$, $\partial\gamma \in \partial X_3$, and $\mathrm{Link}(\partial\Sigma_2, \partial\gamma) = 1$. After the slab compactification, and selecting the physical radius to be $R^2 = \frac{p}{q}$, we get the correct action

$$\mathcal{N}(\partial\Sigma_2)\mathcal{V}_{n,w} \propto \mathcal{V}_{R^2 w, \frac{n}{R^2}}\ . \tag{5.24}$$

Let us make some remarks on the values of $p$ and $q$. If $p, q$ are coprime, the $U$ and $V$ charge lattices overlap only at the identity, and the defect projects out all non-trivial operators (analogous to the irrational $R^2$ in the continuous case). When $p, q$ are such that $\gcd(p,q) > 1$ the charge lattices will overlap at some select values. If $p = q$, the $U, V$ charge lattices overlap for all points such that $C^{\mathbb{Z}_N}(\Sigma_2)$ does not project out any lines and the defect is invertible. This is the discrete analogy of the self-dual radius, requiring $R = pR_0 = \frac{p}{\sqrt{p^2}} = 1$, where the theory is self T-dual.

# Acknowledgments

We are grateful to Andrea Antinucci, Adrien Arbalestrier, Jeremias Aguilera Damia, Christian Copetti, Pierluigi Niro, Giovanni Rizi, Konstantinos Roumpedakis, Pavol Severa, Luigi Tizzano, and Yifan Wang for useful discussions. R.A. and A.C. are respectively a Research Director and a Senior Research Associate of the F.R.S.-FNRS (Belgium). The research of R.A., A.C., G.G. and E.P. is funded through an ARC advanced project, and further supported by IISN-Belgium (convention 4.4503.15). O.H. would like to thank VUB Brussels where most of the work on the project was conducted.

# A  Abelian Gauge Theory Basics

We review the most basic facts about Abelian gauge fields for the sake of clarity. We will be pedantic, as most of the interesting phenomena described in this paper follow from basic

subtleties regarding the quantizaton of periods, large gauge transformations, and related arguments. The main goal of this section is thus to clarify the distinction between the compact and non-compact gauge groups $G = U(1)$ and $\mathbb{R}$, respectively.

A gauge field is a fake one-form. It can in principle only be defined as a one-form on the total space of an associated vector bundle. However, in more down-to-earth terms, it is a one-form up to the transformation

$$A \to A + g^{-1}dg \,. \tag{A.1}$$

where $g : X_3 \to G$ is any function from spacetime into the abelian *group*, not its Lie algebra.

Assuming that the group $G$ is connected (otherwise we can always focus on a connected component), then any element can be written as an exponential of a Lie algebra element as $g = e^{\epsilon T}$, where $\epsilon$ is a parameter, and $T \in \text{Lie}(G)$. However, given a function $g : X_3 \to G$, there may or may not be a corresponding $\epsilon : X_3 \to \text{Lie}(G)$. Put differently, the logarithm $\log(g(x))$ may have branch cuts somewhere in $X_3$. If the log is globally well-defined, we call this a *small gauge transformation*. In this case, we can say that

$$A \to A + d\epsilon \,. \tag{A.2}$$

Otherwise, we say it is a *large gauge transformation*.

For the choice $G = \mathbb{R}$, any $g : X_3 \to \mathbb{R}$ is just a single-valued function on $X_3$, hence a global logarithm can be defined. In this case, all gauge transformations take the form $d\epsilon$, meaning they are all exact one-forms. Therefore, *there are no large gauge transformations* for $G = \mathbb{R}$, regardless of the topology of $X_3$.

For the case $G = U(1)$, the maps $g$ are classified by $\Pi_1(X_3)$, in terms of their winding number. If such a map has non-zero winding, then it does not admit a well-defined global logarithm. The best we can do is write $g(x) = e^{\epsilon(x)}$, for a multi-valued function $\epsilon(x)$. We can still make sense of the expression $g^{-1}dg$, and its period along any loop $\gamma$ will be integrally quantized:

$$\tfrac{1}{2\pi} \int_\gamma g^{-1}dg = \tfrac{1}{2\pi} \int_\gamma d\epsilon \in \mathbb{Z} \,. \tag{A.3}$$

From here, we can draw conclusions about flux quantization. For any Riemann surface $\Sigma$, we can make a cell-decomposition into disks $D_i$ that overlap appropriately. If we focus on one such disk, we can trivialize the connection in its interior, but will be forced to perform a gauge transformation along its boundary $\gamma_i = \partial D_i$ to patch it into the rest of the surface. This gauge transformation, integrated along $\gamma_i$ will be quantized, as we just saw. Using Stokes' theorem, we interpret this as

$$\tfrac{1}{2\pi} \int_{D_i} F = n_i \,, \tag{A.4}$$

the period of the field-strength along the disk. Summing up all such disk integrals, we arrive at the conclusion that

$$\frac{1}{2\pi} \int_\Sigma F = \sum_i n_i \in \mathbb{Z} \,. \tag{A.5}$$

For the case $G = \mathbb{R}$, since there are no large gauge transformations, all such $n_i$ are zero.

To summarize

$$\frac{1}{2\pi} \int_\Sigma F = \left\{ \begin{array}{ll} n & \text{for } G = U(1) \\ 0 & \text{for } G = \mathbb{R} \end{array} \right\} \,. \tag{A.6}$$

Let us now delve into the consequences for Wilson loops. Given a closed curve $\gamma \subset X_3$, we want to define the gauge-invariant observable

$$W_p(\gamma) = \mathrm{e}^{ip \int_\gamma A} \,. \tag{A.7}$$

Under any gauge transformation $A \mapsto A + g^{-1}dg$, we impose that

$$W_p \mapsto W_p \, \mathrm{e}^{ip \int_\gamma g^{-1}dg} \stackrel{!}{=} W_p \,. \tag{A.8}$$

In the $U(1)$ case, this will require $p \in \mathbb{Z}$. In the $\mathbb{R}$ case, any $p$ will do.

To summarize

$$\text{For} \quad W_p(\gamma) \quad p \in \left\{ \begin{array}{ll} \mathbb{Z} & \text{for } G = U(1) \\ \mathbb{R} & \text{for } G = \mathbb{R} \end{array} \right\} \,. \tag{A.9}$$

# B    Details on the Fusion of Condensation Defects

In this section, we provide alternative views and technical details on the computations related to the fusion of condensation defects as discussed in the main text.

## B.1    Defects in the $\mathbb{R}$ BF Theory

We can consider the gauging described by the condensation defects (4.2) from the path integral perspective. This presentation has the advantage of making the fusion of open defects clearer [9].

Let us consider the $C^T_{s=1}(\Sigma)$ defect, defined on a manifold with boundary, $\partial\Sigma \neq 0$. This defect is made from gauging the $\mathbb{R}$ symmetry along the diagonal, i.e.

$$S^{C^T_\Sigma} = \frac{i}{2\pi} \int_\Sigma (a \wedge (b^+ - b^-) + ad\phi) + \frac{i}{2\pi} \int_{\partial\Sigma} (\sigma \wedge (b^+ - b^-) + \sigma d\phi) \,. \tag{B.1}$$

Here $b^\pm$ are the conserved currents for the $\mathbb{R}^{e/m}$ symmetries respectively, $a$ is a 1-form gauge field, $\phi$ is a 0-form field which acts as a Lagrange multiplier to enforce the flatness of $a$, and $\sigma$ is an edge mode that we must introduce to ensure the gauge invariance of the action when $\Sigma$ has a boundary. All these fields take values in $\mathbb{R}$. Their gauge transformations are

$$a \to a + d\lambda^a \quad , \quad \phi \to \phi - \lambda^+ + \lambda^- \quad , \quad \sigma \to \sigma - \lambda^a \, . \tag{B.2}$$

One can check that $S^{C_\Sigma^T}$ in indeed gauge invariant, taking into account that $b^\pm \to b^\pm + d\lambda^\pm$ and that moreover, $b^\pm$ are flat as a result of the bulk equations of motion. Then, the condensation defect can be defined as

$$C_{s=1}^T(\Sigma) = \int Da D\phi D\sigma \, \exp S^{C_\Sigma^T} \, . \tag{B.3}$$

To compute the parallel fusion of the defect $C_{s=1}^T(\Sigma)$ with itself, we add together two copies of (B.1) localised on $\Sigma$ and $\Sigma + \epsilon$ respectively. We must also take into account the non-trivial contribution from the slab of the bulk theory between the two defects. To do so, we consider the equations of motion for the $b^\pm$ fields after gauging on $\Sigma$ and $\Sigma + \epsilon$,

$$db^\pm \pm a\delta(x - x_\Sigma) \pm a'\delta(x - x_{\Sigma+\epsilon}) = 0 \, . \tag{B.4}$$

Since the gauge fields $a, a'$ are imposed to be flat by their equations of motion, we can write

$$b^\pm = \pm a\theta(x - x_\Sigma) \pm a'\theta(x - x_{\Sigma+\epsilon}) \, . \tag{B.5}$$

Plugging this into the bulk and defect actions and taking the limit $\epsilon \to 0$ we find the contribution,

$$\frac{i}{2\pi} \int_\Sigma a \wedge a' \, . \tag{B.6}$$

This contribution is related to the phase factor arising from the linking between the lines making up the two defects. With this, the total action for the fusion becomes

$$\begin{aligned}
S^{(C_\Sigma^T)^2} =& \frac{i}{2\pi} \int_\Sigma \left( (a + a') \wedge (b^+ - b^-) + ad\phi + a'd\phi' + a \wedge a' \right) \\
&+ \frac{i}{2\pi} \int_{\partial\Sigma} \left( (\sigma + \sigma') \wedge (b^+ + b^-) + \sigma d\phi + \sigma' d\phi' \right) \, .
\end{aligned} \tag{B.7}$$

Making the field redefinitions

$$a \to a - a' \, , \qquad \sigma \to \sigma - \sigma' \, , \qquad \phi' \to \phi' + \phi \, , \tag{B.8}$$

the action becomes

$$\begin{aligned}
S^{(C_\Sigma^T)^2} =& \frac{i}{2\pi} \int_\Sigma \left( a \wedge (b^+ - b^-) + ad\phi + a'd\phi' + a \wedge a' \right) \\
&+ \frac{i}{2\pi} \int_{\partial\Sigma} \left( \sigma \wedge (b^+ - b^-) + \sigma d\phi + \sigma' d\phi' \right) \, .
\end{aligned} \tag{B.9}$$

The path integral over $a'$ imposes that $a = -d\phi'$, which implies that all the terms defined on $\Sigma$ are trivialised and we are left with only the boundary terms (after a further shift $\sigma \to \sigma + \phi'$)

$$S^{(C_\Sigma^T)^2} = \frac{i}{2\pi} \int_{\partial\Sigma} \left(\sigma \wedge (b^+ - b^-) + \sigma d\phi + \sigma' d\phi'\right). \tag{B.10}$$

The $\sigma \wedge (b^+ + b^-) + \sigma d\phi$ terms in the action express the gauging of the same $\mathbb{R}$ symmetry along $\partial\Sigma$. The last term, $\sigma' d\phi'$, is a decoupled $\mathbb{R}$ BF theory that will give an (infinite) normalisation factor. Summarizing, we get

$$(C_{s=1}^T(\Sigma))^2 = |H_1(\partial\Sigma, \mathbb{R})| C_{s=1}^T(\partial\Sigma). \tag{B.11}$$

This verifies the non-invertibility of the open defect and the result presented in (4.16), together with the advantage of providing the correct proportionality factor.

We can also consider the fusion of two defects $C_s^T(\Sigma)$ and $C_{s'}^T(\Sigma)$ again from the Lagrangian perspective. Here we take that $\partial\Sigma = 0$. We get

$$S^{C_s^T \times C_{s'}^T} = \frac{i}{2\pi} \int_\Sigma \left(a \wedge (b^+ - sb^-) + ad\phi + a' \wedge (b^+ - s'b^-) + a'd\phi' + \frac{1}{2}(s + s')a \wedge a'\right). \tag{B.12}$$

Here $\frac{1}{2}(s + s')a \wedge a'$ is the linking phase that comes from the contribution of the bulk, see the discussion around (B.6). Now, we can make the field redefinitions

$$\tilde{a} = a + a', \quad \tilde{a}' = -sa - s'a', \quad \tilde{\phi} = \frac{s\phi' - s'\phi}{s - s'}, \quad \tilde{\phi}' = \frac{\phi' - \phi}{s - s'}, \tag{B.13}$$

such that

$$S^{C_s^T \times C_{s'}^T} = \frac{i}{2\pi} \int_\Sigma \left(\tilde{a} \wedge b^+ + \tilde{a}' \wedge b^- + \frac{(s + s')}{2(s - s')}\tilde{a} \wedge \tilde{a}' + \tilde{a}d\tilde{\phi} + \tilde{a}'d\tilde{\phi}'\right). \tag{B.14}$$

The fusion of $C_s^T \times C_{s'}^T$ now expresses the fact that we are gauging the full $\mathbb{R}^e \times \mathbb{R}^m$ symmetry with torsion given by $\tilde{\theta} = \frac{(s+s')}{2(s-s')}$

$$C_s^T \times C_{s'}^T = C^{\tilde{\theta}}. \tag{B.15}$$

Let us explicitly show that this connects with the definition of the $C^\theta$ defect in (4.12). Starting from the Lagrangian (B.14) and choosing $\Sigma = T^2$ for convenience, we can use Poincaré duality to describe the defect as the sum over line insertions along the $\alpha, \beta$ 1-cycles of the torus, i.e.

$$C^\theta(T^2) \propto \int_{a,b,x,y\in\mathbb{R}} e^{2\pi i\tilde{\theta}(xb-ay)} U_y(\beta)U_x(\alpha)V_b(\beta)V_a(\alpha). \tag{B.16}$$

The phase $e^{2\pi i\tilde{\theta}(xb-ya)}$ is the torsion phase originating from the $\tilde{\theta}\tilde{a} \wedge \tilde{a}'$ term in the action. We are not concerned with the normalisation here. By our convention, we wish to resolve

the mesh of lines such that we act on lines along the $\beta$ cycle. This amounts to including the normal ordering phase $e^{\pi i(xb+ya)}$:

$$C^\theta(T^2) \propto \int_{a,b,x,y\in\mathbb{R}} e^{-2\pi i((\tilde{\theta}-\frac{1}{2})ay-(\tilde{\theta}+\frac{1}{2})xb)} U_y(\beta)V_b(\beta)U_x(\alpha)V_a(\alpha) \tag{B.17}$$

Defining $\theta = \tilde{\theta} - \frac{1}{2}$ we get precisely the $C^\theta$ defect in (4.12). Expressing $\theta = \tilde{\theta} - \frac{1}{2} = \frac{s'+s}{2(s'-s)} - \frac{1}{2}$ in terms of $s, s'$

$$\theta = \frac{1}{s'/s - 1} \tag{B.18}$$

which is exactly as we expected in (4.13).

## B.2  Defects in the $U(1)$ BF Theory

Here we check the invertibility of defects in the discrete case through parallel fusion from the summation over lines perspective.

Let us begin by considering the fusion of $C_k^{\mathbb{Z}_q}(T^2)$ (5.20) with itself,

$$C_k^{\mathbb{Z}_q}(T^2) \times C_k^{\mathbb{Z}_q}(T^2) = \frac{1}{q^2} \sum_{a,b\in\mathbb{Z}_q} \sum_{c,d\in\mathbb{Z}_q} e^{-\frac{2\pi i}{q}pkab} e^{-\frac{2\pi i}{q}pkcd} e^{-\frac{4\pi i}{q}pkcb}$$
$$\times U_{pa}(\beta)V_{-pka}(\beta)U_{pc}(\beta)V_{-pkc}(\beta)U_{pb}(\alpha)V_{-pkb}(\alpha)U_{pd}(\alpha)V_{-pkd}(\alpha). \tag{B.19}$$

We have picked up the phase $e^{-\frac{4\pi i}{q}pkcb}$ from moving all $\alpha$ dependent lines to the r.h.s., which will allow us to fuse like operators. Redefining our variables,

$$e = a + c \qquad f = b + d \tag{B.20}$$

we can rewrite the fusion as,

$$C_k^{\mathbb{Z}_q}(T^2) \times C_k^{\mathbb{Z}_q}(T^2) = \frac{1}{q^2} \sum_{e,f\in\mathbb{Z}_q} \sum_{c,d\in\mathbb{Z}_q} e^{-\frac{2\pi i}{q}pk(ef-ed+fc)}$$
$$\times U_{pe}(\beta)V_{-pke}(\beta)U_{pf}(\alpha)V_{-pkf}(\alpha). \tag{B.21}$$

The sums over $c, d$ give the Kronecker deltas,

$$\frac{1}{q}\sum_{c\in\mathbb{Z}_q} e^{-\frac{2\pi i}{q}cpkf} = \delta(pf = 0 \bmod q) \quad , \quad \frac{1}{q}\sum_{d\in\mathbb{Z}_q} e^{\frac{2\pi i}{q}dpke} = \delta(pe = 0 \bmod q) \tag{B.22}$$

where we have used that $\gcd(k,q) = 1$. When $\gcd(p,q) = g > 1$,

$$C_k^{\mathbb{Z}_q}(T^2) \times C_k^{\mathbb{Z}_q}(T^2) = \sum_{e,f\in\mathbb{Z}_g} U_{\frac{N}{g}e}(\beta)V_{-\frac{N}{g}ke}(\beta)U_{\frac{N}{g}f}(\alpha)V_{-\frac{N}{g}kf}(\alpha)$$
$$= g\, C_0^{\mathbb{Z}_g}. \tag{B.23}$$

Notice that when $\gcd(p,q) = 1$ these delta functions are only satisfied when $f = e = 0$ and thus we find $C_k^{\mathbb{Z}_q}(T^2) \times C_k^{\mathbb{Z}_q}(T^2) = 1$. Namely, this verifies the invertibility of $C_k^{\mathbb{Z}_N}$, which corresponds to the case where $p = 1$ and $q = N$.

Likewise, we can also use parallel fusion arguments to show that the opened defect $C_{k=1}^{\mathbb{Z}_N}(\Sigma_2)$ (5.22), $\partial\Sigma_2 \neq 0$, is non-invertible for generic values of $p,q$. We get

$$
\begin{aligned}
C_{k=1}^{\mathbb{Z}_N}(\Sigma_2) \times C_{k=1}^{\mathbb{Z}_N}(\Sigma_2) = \frac{1}{q^2} \sum_{r,s\in\mathbb{Z}_q} \sum_{\ell,t\in\mathbb{Z}_{\gcd(p,q)}} & \mathrm{e}^{-\frac{2\pi\mathrm{i}}{\gcd(p,q)}\ell r} \, \mathrm{e}^{-\frac{2\pi\mathrm{i}}{\gcd(p,q)}st} \, \mathrm{e}^{-\frac{4\pi\mathrm{i}}{\gcd(p,q)}tr} \\
& \times U_{\mathrm{lcm}(p,q)\ell}(\beta)V_{-\mathrm{lcm}(p,q)\ell}(\beta)U_{\mathrm{lcm}(p,q)t}(\beta)V_{-\mathrm{lcm}(p,q)t}(\beta) \\
& \times U_r(\alpha)V_{-r}(\alpha)U_s(\alpha)V_{-s}(\alpha) \,.
\end{aligned}
\tag{B.24}
$$

Redefining the summation variables,

$$
u = \ell + t \qquad v = r + s
\tag{B.25}
$$

we get

$$
\begin{aligned}
C_{k=1}^{\mathbb{Z}_N}(\Sigma_2) \times C_{k=1}^{\mathbb{Z}_N}(\Sigma_2) = \frac{1}{q^2} \sum_{v,s\in\mathbb{Z}_q} \sum_{u,t\in\mathbb{Z}_{\gcd(p,q)}} & \mathrm{e}^{-\frac{2\pi\mathrm{i}}{\gcd(p,q)}(uv+tv-su)} \\
& \times U_{\mathrm{lcm}(p,q)u}(\beta)V_{-\mathrm{lcm}(p,q)u}(\beta)U_v(\alpha)V_{-v}(\alpha) \,.
\end{aligned}
\tag{B.26}
$$

The sums over $s$ and $t$ produce the respective Kronecker delta functions,

$$
\begin{aligned}
\sum_{s\in\mathbb{Z}_q} \mathrm{e}^{-\frac{2\pi\mathrm{i}}{\gcd(p,q)}su} &= q \, \delta(u = 0 \bmod \gcd(p,q)) \\
\sum_{t\in\mathbb{Z}_{\gcd(p,q)}} \mathrm{e}^{-\frac{2\pi\mathrm{i}}{\gcd(p,q)}tv} &= \gcd(p,q) \, \delta(v = 0 \bmod \gcd(p,q)) \,.
\end{aligned}
\tag{B.27}
$$

The fusion becomes

$$
C_{k=1}^{\mathbb{Z}_N}(\Sigma_2) \times C_{k=1}^{\mathbb{Z}_N}(\Sigma_2) = \frac{\gcd(p,q)}{q} \sum_{v=0}^{q-1} \delta(v = 0 \bmod \gcd(p,q)) \, U_v(\alpha)V_{-v}(\alpha) \,.
\tag{B.28}
$$

For general $p,q$ values the r.h.s. does not equal one and therefore $C_k^{\mathbb{Z}_N}(\Sigma_2)$ is non-invertible. However, when $p = q$ the normalization factor is one and the delta function is only satisfied for $v = 0$; everything on the r.h.s. trivializes and the fusion is equal to one. This verifies the expected invertibility of the open defect when the 2d theory is at the self-dual radius.

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
