# Peer review of "Non-Invertible T-duality at Any Radius via Non-Compact SymTFT"

_SciPost Physics_

## Round 1 · Referee Report · Anonymous (Referee 1) · 2025-1-10

Report

This paper provides a clear and explicit discussion of T duality symmetry in the 2d compact boson from a symmetry TFT perspective, identifying the T duality symmetry defect with an open condensation defect of the bulk theory. It gives a nice application of the recent work on symmetry TFT for continuous symmetries. I recommend this paper for publication, though I would like the authors to answer/comment on the following: 1) Can you compare/contrast the approach here for mapping between boundary defects and bulk condensation defects to the non-flat gauging in [28]? 2) Under Eq 4.1, why does this set the normalization "more generally"? A condensation defect can also annihilate lines entirely. Maybe you need it to map at least the trivial line to the trivial line with unit prefactor? 2) What would the integral i.e. in eq 4.2 mean for a concrete operator acting on the Hilbert space in a lattice model? 3) Would there be subtleties with the T duality symmetry defect seeming to have infinite quantum dimension? 4) In a Tambara-Yamagami fusion category there is also a choice of symmetric bicharacter (which in this simple example may be fixed) and Frobenius-Schur indicator. Perhaps you can mention how to incorporate especially the latter piece of data into the symmetry TFT.

Recommendation

Ask for minor revision

  • validity: high
  • significance: high
  • originality: good
  • clarity: top
  • formatting: excellent
  • grammar: perfect

Author:  Riccardo Argurio  on 2025-01-29  [id 5159]

(in reply to Report 1 on 2025-01-10)
Category:
answer to question

We thank both referees for the interesting comments. We post below our replies, following the list of questions of referee 1. They address also the questions of referee 2.

1) Our approach is to use the topological theory in d+1 dimensions, where all the symmetries of the boundary theory have a bulk origin. We further show that also the dualities comes for free once one includes the two U(1) symmetries of the compact boson. We are not sure if the same conclusion can be reached in the same way following the strategy of [28] but it would be interesting to further analyse this path. 2) In section 4 we just studied condensation defects implementing 0-form symmetries of the TFT. In this case the normalization is chosen such that the action is invertible, i.e. it maps a simple line, to a simple line. We have changed a little bit the sentence to avoid stronger claims. 2') The integral is done over the elements of the continuous symmetry \mathbb{R}. Therefore the same expression would be valid in a lattice theory with a continuous global symmetry. 3) Indeed we agree with the referee that the quantum dimension of these defect is infinite in some cases. We think this feature is physical: it is probably related to the fact that the twisted Hilbert space (twisted by the defect) is infinite dimensional, but a more systematic analysis is needed to probe this claim. Notice that, regardless the infinte quantum dimensions, the selection rules implied by these defects are still valid, since quantum dimensions do not enter in the constraints on correlation functions. 4) The extra data of this (generalized) TY symmetry are briefly commented in footnote 19. We believe indeed that the same data characterizing the standard TY category for an abelian finite symmetry are valid to characterize this generalized TY symmetry.

---

## Round 1 · Referee Report · Anonymous (Referee 2) · 2025-1-19

Report

The manuscript discussed T duality in 1+1d compact boson using 2+1d bulk TQFT R x R gauge theory. The T duality is expressed as a condensation defect of the R x R 1-form symmetry. The manuscript is a nice addition to the literature, and I recommend it for publication after the following comments/questions are addressed:

  • In (4.3) the author said they choose a convenient normalization that ensures the fusion is invertible. Can the author clarify how they derive the normalization?

  • In the integral over infinite many lines, can the author comment on what's the measure?

Recommendation

Ask for minor revision

---

## Editorial Decision

resubmitted